# Kombucha Tea-associated microbes remodel host metabolic pathways to suppress lipid accumulation

Rachel N. DuMez-Kornegay[1], Lillian S. Baker[2], Alexis J. Morris[2], Whitney L. M. DeLoach[2], Robert H. Dowen[1,2,3,4]*

1 Curriculum in Genetics and Molecular Biology, University of North Carolina at Chapel Hill, Chapel Hill, North Carolina, United States of America, 2 Department of Biology, The University of North Carolina at Chapel Hill, Chapel Hill, North Carolina, United States of America, 3 Department of Cell Biology and Physiology, The University of North Carolina at Chapel Hill, Chapel Hill, North Carolina, United States of America, 4 Integrative Program for Biological and Genome Sciences, The University of North Carolina at Chapel Hill, Chapel Hill, North Carolina, United States of America

* dowen@email.unc.edu

**Data Availability Statement:** All relevant data are within the paper and its Supporting Information files. The whole genome sequencing data are available at the Sequencing Read Archive

## Abstract

The popularity of the ancient, probiotic-rich beverage Kombucha Tea (KT) has surged in part due to its purported health benefits, which include protection against metabolic diseases; however, these claims have not been rigorously tested and the mechanisms underlying host response to the probiotics in KT are unknown. Here, we establish a reproducible method to maintain *C. elegans* on a diet exclusively consisting of Kombucha Tea-associated microbes (KTM), which mirrors the microbial community found in the fermenting culture. KT microbes robustly colonize the gut of KTM-fed animals and confer normal development and fecundity. Intriguingly, animals consuming KTMs display a marked reduction in total lipid stores and lipid droplet size. We find that the reduced fat accumulation phenotype is not due to impaired nutrient absorption, but rather it is sustained by a programed metabolic response in the intestine of the host. KTM consumption triggers widespread transcriptional changes within core lipid metabolism pathways, including upregulation of a suite of lysosomal lipase genes that are induced during lipophagy. The elevated lysosomal lipase activity, coupled with a decrease in lipid droplet biogenesis, is partially required for the reduction in host lipid content. We propose that KTM consumption stimulates a fasting-like response in the *C. elegans* intestine by rewiring transcriptional programs to promote lipid utilization. Our results provide mechanistic insight into how the probiotics in Kombucha Tea reshape host metabolism and how this popular beverage may impact human metabolism.

## Author summary

Kombucha is a popular fermented tea that has been purported to have many human health benefits, including protection against metabolic diseases like diabetes and obesity. These health benefits are thought to be conferred by the probiotic microbes found in Kombucha Tea, which includes both bacterial and yeast species, that may be able to

(PRJNA1044129). Raw and processed mRNA-Seq data have been deposited in GEO (GSE236037).

**Funding:** This work was supported by NIGMS grant T32GM007092 to R.N.D., NCCIH grant F31AT012138 to R.N.D., and NIGMS grant R35GM137985 to R.H.D. The funders had no role in study design, data collection and analysis, decision to publish, or preparation of the manuscript.

**Competing interests:** The authors have declared that no competing interests exist.

colonize the human intestine and alter host physiology. The mechanisms by which the Kombucha Tea-associated probiotic microorganisms (KTMs) impact host physiology are largely unknown. Using the nematode *Caenorhabditis elegans* as an animal model system to study the host physiological response to KTMs, we show that KTMs colonize the *C. elegans* intestine and impart widespread changes in the expression of evolutionarily conserved lipid metabolism genes, resulting in reduced fat levels in the host. The host metabolic response to actively fermenting KTMs requires an increase in proteins that break down lipids paired with a reduction in a protein that builds triglycerides, which mirrors the events that occur during fasting. These findings are consistent with the reported human health benefits of Kombucha Tea and provide new insights into the host response to Kombucha-associated microbes, which could inform the use of Kombucha in complementary health care approaches in the future.

## Introduction

Since the discovery of antibiotics, humans have been successfully eliminating microbes to cure infections and sterilize our environments, but this nonspecific approach to eliminate pathogenic microbes has also made it increasingly evident just how much we rely on interactions with commensal microbes to remain healthy. Antibiotic use, western diets, a sedentary lifestyle, and many disease states can trigger dysbiosis, or a reduction in microbial diversity, which has been linked to metabolic syndromes, chronic inflammation, and mental health disorders [1–3]. For example, *C. difficile* colitis can arise from antibiotic use and a subsequent loss of microbial diversity in the gut, resulting in severe gastrointestinal symptoms and potentially death [4]. Consumption of probiotics, or live microbes associated with health benefits, can promote, or maintain, a healthy gut microbiome while supplying the host with crucial microbially-derived metabolites [5–7]. Understanding the molecular mechanisms underlying the host response to microbes, particularly probiotics [8], is critical for their incorporation into complementary health care approaches.

Kombucha tea (KT) is a semi-sweet, fermented beverage that is widely consumed as a functional food (*i.e.*, providing health benefits beyond nutritional value) and contains probiotic microbes that have been purported to confer health benefits, including lowering blood pressure, protection against metabolic disease, improved hepatoprotective activity (*i.e.*, protection against liver toxins), and anticancer effects [9–13]. These probiotic microbes include members of the *Acetobacter*, *Lactobacillus*, and *Komagataeibacter* genera [14,15]. While some of these health benefits have begun to be tested in animal models, including the ability of KT to ameliorate diabetic symptoms or limit weight gain in adult mice [16–19], the mechanistic underpinnings of these phenotypes have not been rigorously investigated. Moreover, the interactions between the microbes in Kombucha Tea, which include both bacterial and yeast species, and the host remain completely unexplored. Because this beverage contains live probiotic microbes and is widely consumed under the largely unsubstantiated claim that it confers health benefits, it is imperative to gain mechanistic insight into the host physiological and cellular response to KT consumption.

The impact of individual probiotic microbes, or in this case the small community of Kombucha-associated microbes, on human physiology is difficult to deconvolute as humans consume a complex diet, have trillions of microbes colonizing their gut, and mechanistic investigation of host-microbe interactions is not feasible in human subjects. Therefore, use of animal models is essential to investigate how probiotic consumption influences host physiological processes. *Caenorhabditis elegans* has been widely used to investigate mechanisms of metabolic regulation and how nutrient sensing pathways govern organismal homeostasis

[20,21]. *C. elegans* is also an emerging model for studying the impact of the gut microbiome on host physiology [22,23]. Axenic preparation of *C. elegans* cultures renders these bacterivore animals microbe-free at the onset of life, allowing for complete experimental control over which microbes are consumed during their lifetime (*i.e.*, animals are germ-free before encountering their microbial food source). Additionally, microbes that escape mechanical disruption during feeding can robustly colonize the intestinal lumen [22–24]. Thus, the simple digestive tract of *C. elegans* is effectively colonized by bacteria that are provided as a food source, making it an ideal system to interrogate the host metabolic response to consumption of specific microbes. Indeed, previous studies have successfully used *C. elegans* to investigate how individual species of microbes, including probiotics, can elicit physiological changes by rewiring conserved genetic pathways [25–31].

Here, we use *C. elegans* to investigate whether intestinal colonization with Kombucha-associated microbial species (KT microbes or KTMs) rewires host metabolism. We developed a reproducible method to culture animals on lawns of KT microbes consisting of microbes found in all commercial and homebrewed KTs (*i.e.*, bacteria from the *Acetobacter* and *Komagataeibacter* genera and a yeast species). We found that animals feeding *ad libitum* on KT microbes accumulate significantly less fat than animals consuming either an *E. coli* diet, any of the individual three KT-associated microbial species, or a simple non-fermenting mix of these three species. Furthermore, our data suggest that KT consumption reduces fat storage by modulating host lipid metabolism pathways rather than restricting caloric intake. To gain insight into the mechanisms that underlie this reduction in lipid levels, we performed a transcriptomic analysis of KT microbe-fed animals, which revealed that a class of lysosomal lipases that function in lipophagy was up-regulated and that a crucial enzyme in triglyceride synthesis was down-regulated in response to KT microbes. Our results suggest that Kombucha Tea consumption may alter lipid droplet dynamics by promoting their degradation via lipophagy, while simultaneously restricting lipid droplet expansion through down-regulation of triglyceride synthesis. This investigation lays crucial groundwork to deconvolute the molecular mechanisms that may underlie the purported health benefits of KT using a genetically tractable animal model.

## Results

### Rearing *C. elegans* on a lawn of KT microbes results in reproducible colonization of the gut

Small batch brewing of KT is a serial fermentation process in which the microbially-generated biofilm and a small amount of fully fermented liquid culture are transferred to a fresh preparation of sucrose media, which then ferments for at least a week prior to consumption. This traditional method of brewing KT results in a dynamic microbial community and pH shift over the course of fermentation (pH decreases from 7 to ~4). Contamination by environmental microbes is limited since these species are outcompeted by the core KT microbes (KTMs) as the pH drops [13,32–34]. Furthermore, construction of the protective pellicular biofilm, colloquially referred to as a SCOBY (symbiotic culture of bacteria and yeast), by the KTMs reduces outside contamination [35]. To investigate the physiological and metabolic effects of Kombucha Tea consumption using a genetic model system, we first sought to establish a reproducible method to deliver KTMs to *C. elegans* animals via feeding on our standard agar-based nematode growth media (NGM), which do not contain any antibiotics or antifungals. We found that seeding KTMs that are actively growing in a KT homebrew onto NGM plates is sufficient to generate a lawn of microbes that expands in population and produces a biofilm over the course of 4 days (S1A and S1B Fig).

To gain a better understanding of the microbial community dynamics in our KT culture and to assess our ability to recapitulate the KT microbial community on NGM plates, we performed 16S rDNA sequencing of the fermenting KT culture and the KTMs washed from NGM plates isolated from three representative brew cycles (S1 Table). After six days of fermentation, the microbial communities in the culture and on NGM plates were similar and were dominated by the expected set of Kombucha-associated microbes (*i.e.*, *Acetobacter* and *Komagataeibacter* species), which are essential components of all commercial or homebrewed KTs (Figs 1A, 1B and S1C–S1E) [36]. Notably, the KT culture microbial community remained similar through day 12 of fermentation; however, the community on NGM plates was no longer dominated by the expected KTMs at day 12, which may be due to the expansion of environmental microbes (Fig 1A and 1B). Thus, we exclusively used NGM plates between days 4–8 after KTM seeding for our subsequent experiments. Establishing this method to reproducibly culture the KT microbial community on NGM plates was essential to leveraging *C. elegans* as a model to study the host response to KT consumption.

Using this standardized method of KTM culturing, we next sought to determine if populations of *C. elegans* could be reared on a diet exclusively consisting of KTMs. Given that KTMs are a mix of microbial and yeast species, we first conducted an avoidance assay to assess whether *C. elegans* animals would remain on or flee from the lawn, which is a typical response to a pathogen [37–40]. Importantly, animals remained on the KTM microbial lawn throughout development and into adulthood at levels similar to the *E. coli* controls (Fig 1C and 1D), indicating that *C. elegans* animals can be successfully reared on KT microbes. These comparisons, as well as all our subsequent characterizations of the KTMs, were conducted in conjunction with two standard laboratory *E. coli* diets (OP50 and HT115 [41]) and a strain of the bacterium *Acetobacter tropicalis* that we isolated from our KT culture. *A. tropicalis* is a major constituent of all KTs and produces vitamin B12 among other bioactive molecules found in KTs [14,15,34,35,42–45]. OP50 and HT115 *E. coli* strains modulate *C. elegans* physiology differently, which can be partially attributed to differences in vitamin B12 levels [28,46–48]. Interestingly, when presented with a choice of diets animals did not select the KTM lawn (S2A and S2B Fig). This behavior was consistent across different *C. elegans* wild isolates and other *Caenorhabditis* species (S2C–S2I Fig), suggesting that these animals are either attracted to the *E. coli* or are repelled by a component of the KTM culture. Though animals seem to prefer other food sources, animals offered only KTMs do not flee the lawn, demonstrating that *C. elegans* can be reliably reared on a KTM-exclusive diet using standard *ad libitum* feeding methods.

To test whether KTMs altered feeding behavior, which might result in reduced caloric intake, we measured pumping rates (*i.e.*, the rate at which animals' pharyngeal muscle contracts to intake food) of individual animals consuming KTMs or control food sources. We found no significant difference in pumping rates of animals consuming KTMs compared to any of the other food sources (Fig 1E), suggesting feeding behavior is not altered on KTM lawns. Finally, we assessed whether KTMs colonize the intestinal lumen of *C. elegans* animals, as would be predicted for these probiotic microbes in the human gastrointestinal tract. After rearing animals on different diets, we removed surface microbes, extracted the intestinal microbes, and quantified the colony forming units (CFUs) present. Animals consuming KTMs contained at least 5 times more CFUs than animals consuming any other diet, indicating that KTMs robustly colonize the *C. elegans* gut (Fig 1F). To further investigate this intestinal colonization, we used scanning electron microscopy to image the intestine of animals consuming KTMs and found intact microbial cells present in the intestinal lumen (Fig 1G). Together, these results demonstrate that *C. elegans* animals can be successfully reared on a KTM-exclusive diet, which closely mirrors the microbial community found in the KT culture, resulting in robust KTM colonization of the gut.

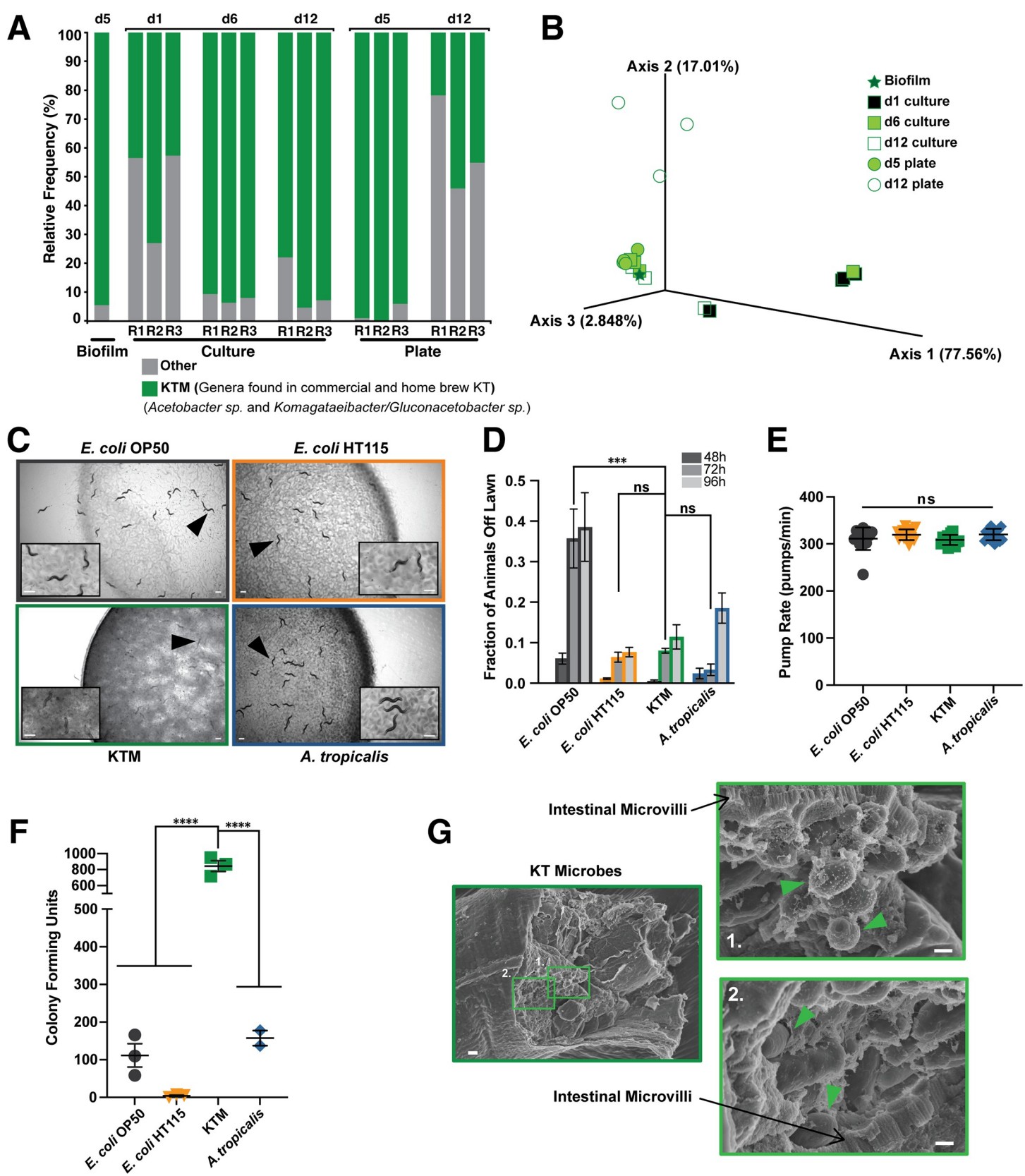

**Fig 1. Consumption of Kombucha-associated microbes (KTMs) does not impact feeding behavior and results in robust gut colonization of *C. elegans* animals.** (A) 16S rDNA sequencing of traditionally cultured KTMs, the biofilm of an actively fermenting Kombucha Tea, or KTMs grown on NGM agar plates. The frequency of known KTMs (green) is plotted relative to environmental microbial contaminants (gray) for three biological replicates across twelve days (d1-d12). (B) A Principal Component Analysis of the weighted unifrac beta diversity derived from the 16S rDNA sequencing revealed similarity between d5 plates and the d6 culture, but divergence of the d12 plates. (C) Images (scale bar, 500 μm) and (D) quantification (mean ± SEM) of day 1 adult animals on lawns of the indicated microbes (72 hr timepoint compared, ***, $P<0.001$, ns, not significant, one-way ANOVA). (E) Measurements of pumping rates for day 1 adults consuming each microbial food source (mean ± SD, ns, not significant, one-way ANOVA). (F) Quantification of the microbial CFUs from animals consuming each diet shows KTMs colonize the gut at higher levels compared to the control diets (mean ± SEM, ****, $P<0.0001$, one-way ANOVA, three biological replicates, 10 animals per replicate). (G) Representative scanning electron microscopy images of KTMs in the intestinal lumen (black arrows point to the intestinal microvilli and green arrow heads indicate intact KTMs; left scale bar, 2 μm; right scale bars, 1 μm). Expanded data for panel A can be found in S1 Table and the raw data underlying panels D, E, and F can be found in S1 Data.

## Animals consuming Kombucha microbes exhibit reduced fat accumulation

Dietary components, including those produced by probiotic microbes, can play a substantial role in modulating host metabolism, including lipid storage and lipolysis [49–51]. Consistently, *C. elegans* metabolism is remarkably sensitive to differences in microbial diets, as even highly similar strains of *E. coli* promote markedly different levels of fat content [28,29,41]. Given the purported metabolic benefits of KT in humans, including decreased risk of obesity [9–13], we reasoned that consumption of KTMs may impact lipid levels in *C. elegans*. The majority of fat in *C. elegans* animals is stored in intestinal epithelial cells within lipid droplets in the form of triglycerides (TAGs), with smaller lipid deposits found in the hypodermis and germline [52]. Using the well-established lipophilic dyes Oil Red O and Nile Red, which both stain neutral lipids, we examined the fat content of animals consuming KTMs and control microbes [52,53]. Animals consuming KT microbes accumulated significantly less fat than animals consuming other food sources, including *A. tropicalis*, which is particularly noteworthy given that *A. tropicalis* is the most abundant microbial species in KT (Fig 2A–2D). These trends continued during and after the reproductive period (S3A Fig), suggesting that KTMs restrict host lipid accumulation throughout reproduction and during the aging process. Importantly, the KTM-fed animals successfully commit a significant proportion of their somatic fat stores to the germline and developing embryos at adulthood (Fig 2C), suggesting that reproductive programs are not impaired despite the overall reduction in lipid levels. The decrease in Oil Red O and Nile Red staining suggests that animals consuming KTMs may have reduced TAG levels compared to animals on control diets. Therefore, we used a biochemical assay to quantify the total amount of TAGs in populations of animals fed each diet [54,55]. Consistent with our previous observations, animals consuming KTMs had an ~85% or ~90% decrease in TAG levels compared to animals consuming *E. coli* OP50 or *A. tropicalis*, respectively (Fig 2E). Together, these data clearly demonstrate that animals consuming KT microbes accumulate less fat than *E. coli*-fed animals and that the most abundant microbe in KT, *A. tropicalis*, is not sufficient to recapitulate this phenotype. This finding is particularly relevant to human health, as KT consumption has been shown to restrict weight gain and alleviate diabetic symptoms to a similar degree as metformin in rodent models [16–19].

Given that the major site of lipid storage in *C. elegans* is in intestinal lipid droplets (LD), we hypothesized that LD size or abundance may be impacted in the intestine of KTM-fed animals. Taking advantage of a transgenic strain that expresses the LD-residing DHS-3::GFP protein (*dhs*, dehydrogenase, short chain), we measured LD abundance and size in intestinal cells of animals fed each diet. Both lipid droplet size and abundance were dramatically reduced in animals consuming KTMs relative to *E. coli*- or *A. tropicalis*-fed animals (Figs 2F–2H and S3B). Together, these results suggest that regulation of lipid droplet synthesis or stability may account for the reduced lipid accumulation that we observed in KTM-fed animals.

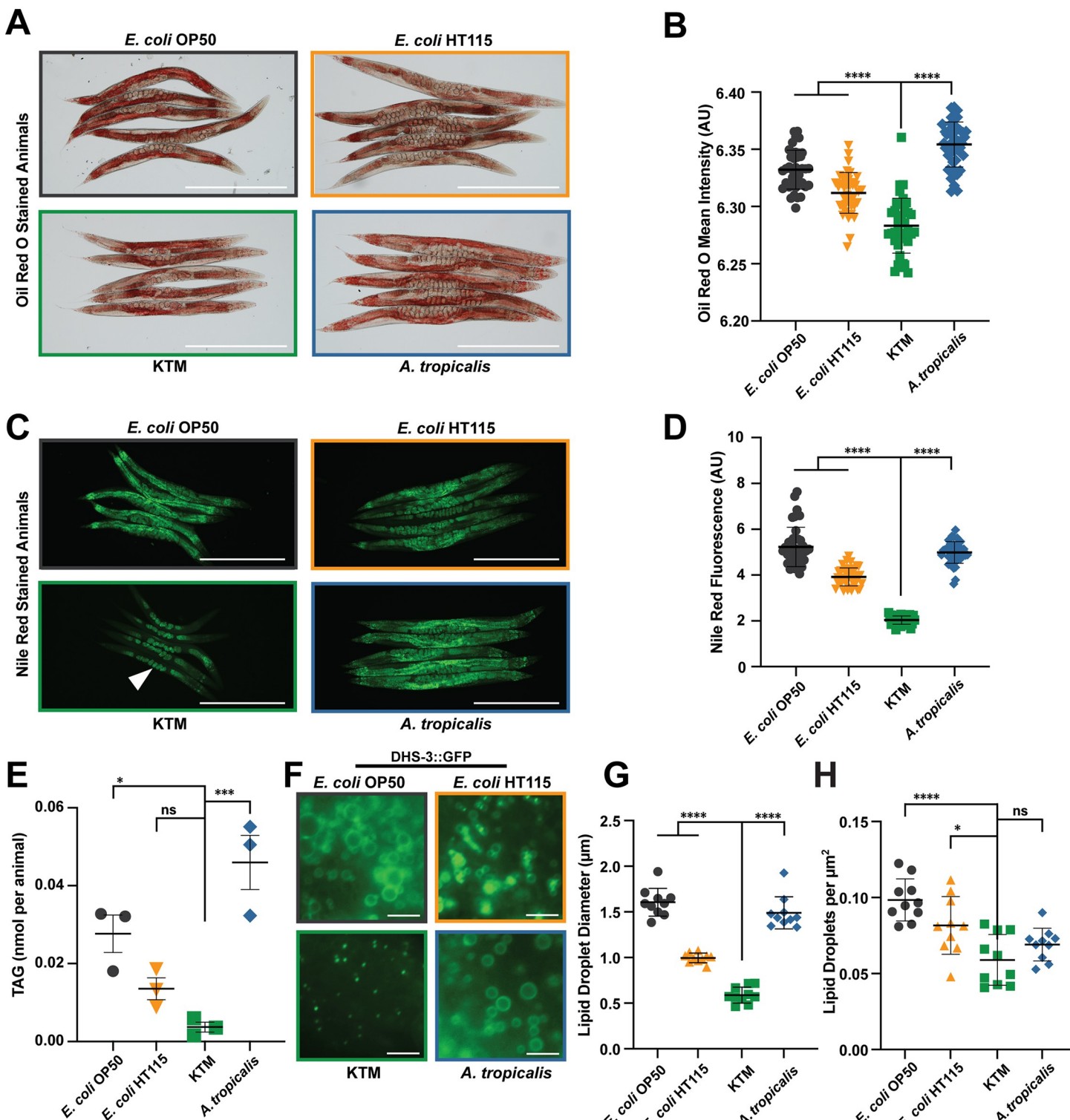

**Fig 2. KTMs restrict lipid accumulation in the host.** (A) Representative images (scale bar, 500 μm) and (B) quantification (mean ± SD, ****, *P*<0.0001, one-way ANOVA) of day 1 adults stained with Oil Red O. (C) Representative fluorescence images (scale bar, 500 μm) and (D) quantification (mean ± SD, ****, *P*<0.0001, one-way ANOVA) of day 1 adults stained with Nile Red. (E) Biochemical quantification of the triglycerides (TAGs per animal) in animals consuming each food source (mean ± SEM, ***, *P*<0.001, *, *P*<0.05, ns, not significant, one-way ANOVA). (F) Representative fluorescence images of DHS-3::GFP (*dhs*, <u>d</u>ehydrogenase, <u>s</u>hort chain) at intestinal lipid droplets in animals consuming the indicated microbial diets (scale bar, 5 μm). (G) Lipid droplet size measurements with each datapoint representing the average intestinal lipid droplet diameter for a single animal (mean ± SD, ****, *P*<0.0001, one-way ANOVA). (H) Lipid droplet density measurements with each datapoint representing the number of lipid droplets per μm² for a single animal (mean ± SD, ****, *P*<0.0001, *, *P*<0.05, ns, not significant, one-way ANOVA). Raw data underlying panels B, D, E, G, and H can be found in S2 Data.

## KTM consumption accelerates growth rates and does not substantially alter fecundity

Different microbial diets can have a profound impact on *C. elegans* growth rate and fecundity [29,30]. A KTM diet could restrict developmental rate or alter reproductive programs. Moreover, reduced nutrient absorption stemming from a KTM diet could result in caloric restriction and reduced lipid accumulation. Indeed, genetic or nutritional models of caloric restriction cause animals to develop more slowly, to accumulate less intestinal fat, and to have a delayed reproductive period that ultimately results in less progeny production [28,56–58].

Therefore, we sought to determine whether animals consuming KTMs exhibit slower developmental rates and smaller brood sizes than animals consuming either an *E. coli* or *A. tropicalis* diet. To investigate variations in developmental rate, we employed a transgenic strain expressing a GFP-PEST protein under the control of the *mlt-10* promoter (P*mlt-10*::*GFP-PEST*), which is specifically expressed during each of the four molt stages, resulting in four peaks of GFP fluorescence throughout development (Fig 3A). The PEST amino acid sequence ensures rapid GFP turnover by proteolytic degradation and allows for precise temporal analyses. Animals consuming KTMs molt at a similar, if not an accelerated rate relative to animals on the control food sources (Fig 3A–3C), clearly indicating that KTM consumption does not decrease developmental rate. To gain a more comprehensive view of animal development during KTM consumption, we performed mRNA sequencing (mRNA-Seq) of adult animals consuming *E. coli*, *A. tropicalis*, or KTMs. Upon inspection of 2,229 genes previously associated with *C. elegans* development [29], we observed very few gene expression differences between KTM-fed animals and those fed control diets (Fig 3D–3F), suggesting that the KTM-fed population reaches adulthood synchronously. Together, these results suggest that animals consuming KT microbes exhibit wild-type development.

Caloric restriction has a profound impact on *C. elegans* physiology, including reduced developmental rate [58]. The *eat-2* mutant is a genetic model of caloric restriction, as loss of *eat-2* results in impaired pharyngeal pumping and reduced nutrient intake [58]. Reducing nutrient availability (*i.e.*, *E. coli* OP50 lawns with concentrations $\leq 10^9$ CFU/ml) provides a second effective method of caloric restriction [59]. Therefore, to further evaluate whether animals consuming KTMs are calorically restricted (CR), we conducted developmental rate assays with wild-type and *eat-2* mutant animals consuming *ad libitum E. coli* lawns, CR *E. coli* lawns ($10^8$–$10^9$ CFU/ml), or our standard *ad libitum* lawns of KTMs. This analysis revealed that both wild-type and *eat-2* animals exhibited accelerated developmental rates when consuming KTMs compared to the *E. coli* OP50 diet (Fig 3G). Importantly, *eat-2* animals showed reduced developmental rates on CR *E. coli* lawns relative to *ad libitum E. coli* lawns, indicating that the effects of the *eat-2* mutation are further enhanced by additional caloric restriction; however, KTM-feeding partially suppressed the developmental defects of the *eat-2* mutation (Fig 3G). These data demonstrate that KTM consumption does not mimic the effects of restricted caloric intake.

Reproductive output (*i.e.*, brood size) of *C. elegans* is modulated by diet, possibly through the tuning of reproductive programs at the transcriptional level [29,60]. Therefore, we measured the brood sizes of animals consuming KTMs and control diets, finding that the average brood size of animals consuming KTMs was only modestly lower than those consuming *E. coli* OP50 (Figs 3H and S4; 295 versus 240, $P < 0.05$). Additionally, we found that animals consuming KTMs lay their eggs at a similar rate relative to *E. coli*-fed animals (Figs 3I and S4). In contrast, calorically restricted animals, such as *eat-2* mutants, have extended egg laying periods, up to 12 days, and have substantially reduced brood sizes, with *eat-2* mutants averaging 100–175 progeny [28,57]. Thus, the ~20% reduction in fertility for KTM-fed animals is inconsistent with the more severe reduction in brood size of CR animals. It could, however, be consistent

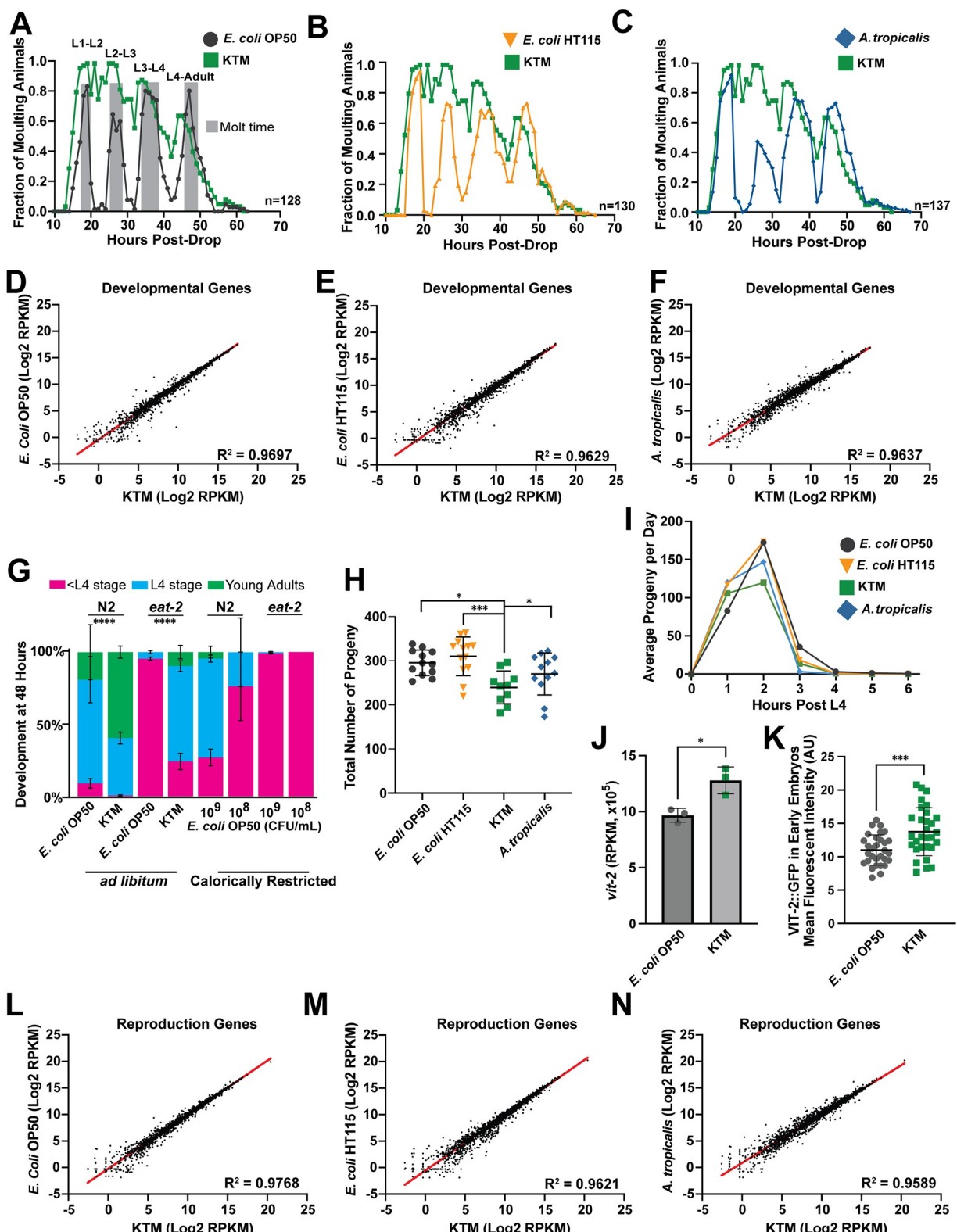

**Fig 3. Developmental timing is accelerated and reproductive output is only modestly reduced during KTM feeding, suggesting that caloric intake is not restricted by KTM consumption.** (A-C) Profiles of P*mlt-10*::*GFP-PEST* expression throughout development after dropping

synchronized L1s on the indicated microbes. The reporter is expressed exclusively during the larval molts (shown in gray in A). A single representative experiment is displayed in panels A-C. (D-F) Scatter plots comparing the expression of 2,229 developmental genes as determined by mRNA-Seq (RPKM, reads per kilobase of transcript per million mapped reads). A linear regression analysis and the corresponding $R^2$ value is reported for each comparison. (G) The frequency (mean ± SEM) of wild-type N2 and *eat-2(ad465)* individuals at the indicated developmental stages after 48 hours of growth on *ad libitum* KTM, *ad libitum* E. coli, or caloric restriction *E. coli* ($10^8$ or $10^9$ CFUs/mL) plates (****, $P<0.0001$, chi-squared test). (H) Brood sizes of wild-type animals reared on the different diets (mean ± SD, ***, $P<0.001$, *, $P<0.05$, one-way ANOVA). (I) A plot of progeny production for each day during the reproductive period demonstrating that KTM-fed animals exhibit a similar egg laying rate compared to *E. coli* OP50-fed animals. (J) Normalized *vit-2* gene expression values (RPKM, reads per kilobase of transcript per million mapped reads; mean ± SEM, *, $P<0.05$, T-test) and (K) quantification of VIT-2::GFP fluorescence in early embryos (mean ± SD, ***, $P<0.001$, T-test) from animals consuming an *E. coli* OP50 or KTM diet. (L-N) Scatter plots and a linear regression analysis ($R^2$ value reported) comparing the expression of 2,367 reproduction genes as determined by mRNA-Seq. Raw data underlying panels A-N can be found in S3 Data.

with impaired maternal provisioning of lipid-rich yolk to oocytes from intestinal fat stores, a process termed vitellogenesis. Thus, we next examined the mRNA levels of *vit-2*, which encodes a vitellogenin protein that mediates the intestine-to-oocyte transport of lipids, finding that *vit-2* levels are increased in animals fed a KTM diet compared to *E. coli*-fed animals (Fig 3J). Consistently, vitellogenin protein levels, which we measured in early embryos (prior to the 44-cell stage) using an endogenously tagged VIT-2::GFP protein, were also elevated in KTM-fed animals (Fig 3K), further substantiating that KTM consumption does not impair maternal lipid provisioning. Finally, we inspected the expression of 2,367 genes implicated in reproduction [29], finding that KTM consumption does not broadly alter reproductive gene expression programs relative to control diets (Fig 3L–3N). Together, our results indicate that reproductive programs are not dramatically altered in animals consuming KTMs. This finding, along with the observation that animals consuming KTMs exhibit wild-type developmental rates, is consistent with the contention that caloric intake is not impaired during KTM consumption and substantiates *C. elegans* as model to investigate the impact of Kombucha-associated microbes on host metabolic pathways.

## Long-term KTM co-culturing is required to remodel host metabolism

Sequencing of commercially available and non-commercial-small-batch KTs has revealed that a reproducible set of core microbes are found in KT [14,15,34,36,61]. These include bacteria in the *Acetobacter*, *Komagataeibacter*, *Gluconacetobacter*, *Gluconobacter*, and *Lactobacillus* genera, as well as yeast in the *Brettanomyces*, *Zygosaccharomyces*, *Candida*, *Dekkera*, *Lachancea*, and *Schizosaccharomyces* genera. Furthermore, Huang and colleagues recently established a minimal KT microbiome that recapitulates key aspects of traditionally brewed KT based on the criteria that this microbial mix could (1) coexist as in KT, (2) produce a KT-like biochemical composition, and (3) build a pellicle. Intriguingly, regardless of the ratio of bacteria-to-yeast at the onset of fermentation, by day 6 this ratio stabilizes with relatively equal representation of each species regardless of the concentration of the microbial species combined [36].

Given that KTMs robustly colonize the *C. elegans* gut and that feeding animals the known dominant KT microbe, *A. tropicalis*, fails to recapitulate the host response to KTMs, we sought to identify additional microbes from our KT culture that can colonize the intestine of animals after KTM consumption. Isolation of these species would facilitate the creation of a KTM culture consisting of a minimal microbiome core, which may be sufficient to confer metabolic phenotypes in *C. elegans* animals. Our initial extraction of intestinal microbes from KTM-fed animals (Fig 1F) isolated a bacterial species, *Acetobacter tropicalis*, and a yeast species, of either the *Zygosaccharomyces* or *Brettanomyces* genera, which we identified by 16S and 18S rDNA sequencing, respectively (S5A and S5B Fig). While these microbes represent two of the species commonly found in KT, they do not constitute a minimal KT culture because they cannot form a pellicle [36]. Therefore, we sought to isolate the cellulose-producing species from our culture that is responsible for building the pellicle. We removed a small piece of the biofilm

from our KT culture and used a combination of enzymatic digestion (driselase) and mechanical disruption (sonication) to dislodge the bacteria from the cellulose matrix. The cellulose-producing bacterium was isolated on mannitol agar plates containing Calcofluor White, which stains cellulose and chitin and fluoresces under ultraviolet light [62–64]. This strategy resulted in the isolation of an additional KT microbe that was identified as a member of the *Komagataeibacter* genus by 16S rDNA sequencing (S5C Fig).

To gain additional genetic information about our individually isolated KT microbes, we performed short read whole genome sequencing of the genomic DNA. Subsequently, the Kraken algorithm [65], a bioinformatic pipeline for metagenomic classification, was used to determine the approximate taxonomy of our individual KT microbes. Based on these taxonomical classifications, as well as a compiled list of previously published KT-associated microbes, we aligned our KT microbe sequences to several available reference genomes to gain species level information (Fig 4A and S2 Table). This strategy identified our KT microbes as *Komagataeibacter rhaeticus* (98.76.% alignment rate to strain ENS_9a1a), *Acetobacter tropicalis* (87.55% alignment rate to strain NBRC101654), and *Zygosaccharomyces bailli* (86.88% alignment rate to strain CLIB213) [66].

The isolation and identification of the dominant KT microbes from our culture allowed us to further investigate how consumption of the individual KT microbes, or mixtures of microbes, alter *C. elegans* lipid metabolism (Fig 4B). We initially fed the individual KT microbes to animals, finding that diets of *A. tropicalis* or *K. rhaeticus* promoted lipid accumulation at levels similar to *E. coli*-fed animals, while a diet of the yeast species *Z. bailli* failed to support animal development (Figs 4C and S6A–S6D). Surprisingly, increasing the concentration of KTMs present in the lawn fivefold (5x KTM) further reduced lipid levels compared to our standard KTM lawn (S6A–S6C Fig), indicating that an increase in the microbial concentration, which likely results in additional available nutrients, does not increase host lipid accumulation.

We then hypothesized that a mixture of *K. rhaeticus*, *Z. bailii*, and *A. tropicalis* would represent the minimal core of KT microbes, which when co-cultured would ferment sucrose, build a pellicle, and produce a biochemical composition similar to Kombucha tea. Therefore, we combined the three KT microbe isolates in filter sterilized KT media (*i.e.*, steeped black and green tea containing ~5% sucrose) and allowed them to ferment for several weeks until a pellicle was formed. We refer to this *de novo* KT as KTM-Fermented Mix or "KTM-FM" (Fig 4B). To assess the ability of our KTM-FM culture to alter host lipid metabolism, we performed Oil Red O staining on animals consuming KTM-FM or a simple non-fermenting mix of the three KT microbes (referred to as KTM-Mix, abbreviated "KTM-M", Fig 4B). Intriguingly, we found that the KTM-M diet did not reduce lipid accumulation, lipid droplet size, or lipid droplet abundance in the host (Figs 4C–4E and S6E); however, consumption of KTM-FM reduced lipid levels to a similar degree as the original KTM diet (Fig 4F). Importantly, neither the KTM-M nor the KTM-FM diet impaired developmental or behavioral programs (S6F–S6I Fig). These results suggest that long-term fermentation is necessary for the host metabolic response to KT consumption. Furthermore, the observation that a non-fermented mix of KT microbes fails to restrict host lipid accumulation further supports our conclusion that KTM-fed animals are not calorically restricted.

To better understand the importance of fermentation time, we fed animals KTM-FM cultures after different lengths of fermentation and measured lipid levels using Oil Red O staining. Animals that were fed KTM-FMs with fermentation times less than one week had elevated lipid levels; however, KTM-FMs fermented for 2 weeks or more promoted the depletion of host lipids (Fig 4G). Additionally, removal of the fully fermented KTM supernatant followed by repeated washes of the KTMs with a 5% sucrose solution prior to seeding the NGM plates did not alter host lipid accumulation in response to KTMs (Fig 4H and 4I), suggesting that the small molecules

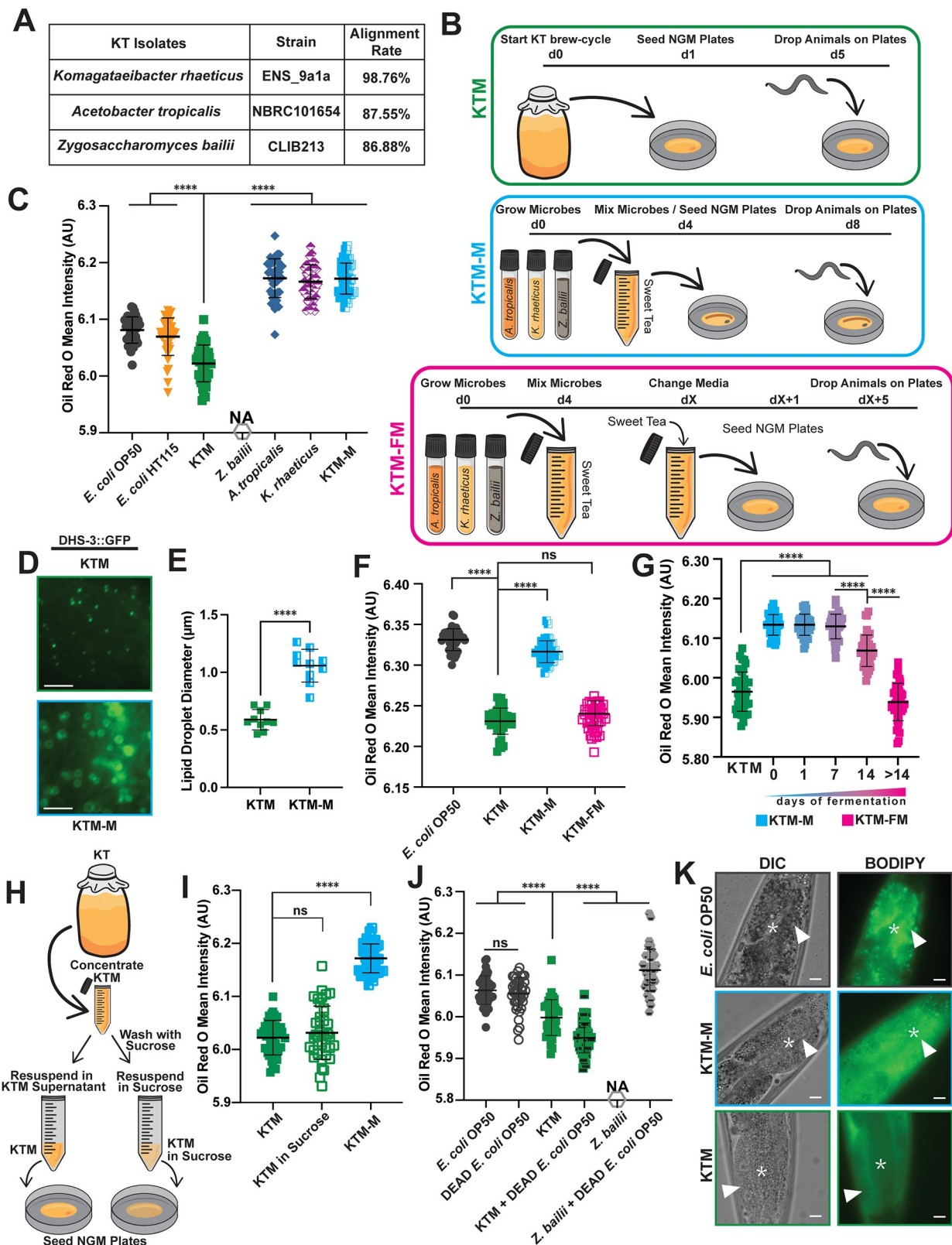

**Fig 4. The host lipid depletion response is specific to actively fermenting KTM cultures and is not conferred by individual microbes or a non-fermenting mixture.** (A) Purification and whole genome sequencing of the microbes from our Kombucha culture resulted in species-level

identification of the core KTMs. (B) A schematic of the preparation and delivery methods for the three KT-derived diets (orange, *A. tropicalis*; tan, *K. rhaeticus*; gray, *Z. bailii*; d, days). Sweet tea media consists of a black and green tea mix with 5% cane sugar that has been filter sterilized. KTM cultures are maintained via serial fermentation, while the KTM-M and KTM-FM are *de novo* cultures. (C) Quantification of Oil Red O staining of day 1 adults fed the indicated diets (mean ± SD, ****, $P<0.0001$, one-way ANOVA). A *Z. bailii* diet does not support animal development. (D) Representative fluorescence images (scale bars, 5 μm) and (E) quantification of lipid droplet diameter (mean ± SD, n = 10 individuals, ****, $P<0.0001$, unpaired T-test) in the intestines of DHS-3::GFP transgenic animals. The KTM lipid droplet image and size measurements shown in (D-E) are also displayed in Fig 2F and 2G, as all these samples were processed in parallel. (F-G) Quantification of Oil Red O staining of day 1 adults fed the indicated KT diets (mean ± SD, ****, $P<0.0001$, ns, not significant, one-way ANOVA). (F) Animals consuming KTM-FM have similar lipid levels as KTM-fed animals, while (G) KTM-M must be co-cultured for at least 14 days to restrict host lipid accumulation. (H) The experimental design to test whether the KTM culture supernatant is required for host lipid depletion. (I) Quantification of Oil Red O staining of day 1 adults following a diet of KTMs, KTMs washed extensively with 5% sucrose, or KTM-M (mean ± SD, ****, $P<0.0001$, ns, not significant, one-way ANOVA). (J) Quantification of Oil Red O staining of day 1 adult animals consuming *E. coli*, KTMs, and *Z. bailii* with or without dead *E. coli* supplementation as an inert nutrient source (mean ± SD, ****, $P<0.0001$, ns, not significant, one-way ANOVA). (K) Representative DIC and fluorescence images of the intestine after feeding the indicated diets supplemented with C1-BODIPY-C12 (stars indicate the intestinal lumen and arrowheads indicate the intestinal epithelial cells; scale bars, 10 μm). Expanded data for panel A can be found in S2 Table and the raw data underlying panels C, E, F, G, I, and J can be found in S4 Data.

in the green and black tea may be dispensable for conferring host lipid phenotypes. This result, however, does not rule out the possibility that the tea-derived metabolites are crucial for establishing the symbiotic Kombucha culture. Together, these data argue that KT microbes must form an established community to reconfigure host lipid metabolism pathways.

Although we observed colonization of *C. elegans* gut with *A. tropicalis* (Fig 1F), it is unclear whether the other KT isolates, *K. rhaeticus* or *Z. bailii*, are ingested by animals. To visualize these microbes in the gut of live animals, we stained animals fed *E. coli*, *K. rhaeticus*, or KTMs with Calcofluor White, which selectively stains the polysaccharides in chitin and cellulose. We observed the cellulose-producing microbe *K. rhaeticus*, which supports animal development, in the intestinal lumen (S6J Fig), suggesting that *K. rhaeticus* bacteria can colonize the gut while synthesizing cellulose. Surprisingly, we also observed chitin-producing yeast cells in the intestinal lumen, indicating that *Z. bailii* can be consumed by animals at the adult stage (S6K and S6L Fig). Importantly, these results are consistent with the presumption that all three of KT microbes (*Z. bailii*, *K. rhaeticus*, and *A. tropicalis*) isolated from our KT culture can escape mechanical disruption in the pharynx and can be found in the intestinal lumen of *C. elegans*. To further assess the ability of the KT microbes to colonize the gut, we quantified the intestinal lumen size of animals reared on the *E. coli* OP50, KTM, and KTM-M diets. Using animals expressing ERM-1::GFP, which localizes to the apical surface of intestinal cells and facilitates luminal measurements, we found that individuals consuming a KTM diet had an increased intestinal lumen diameter compared to animals consuming *E. coli* OP50 but not the KTM-M diet, suggesting that any diet containing KT microbes stimulates intestinal bloating (S6M Fig).

The presence of *Z. bailii* in the gut, which may contribute to intestinal bloating, raised the possibly that the yeast (or the other KT microbes) may restrict nutrient absorption, resulting in caloric restriction. Therefore, we supplemented KTM and *Z. bailii* diets with heat killed *E. coli* OP50 and assessed lipid levels using Oil Red O staining. While *E. coli* supplementation had little impact on the ability of the KTM diet to restrict host lipid accumulation, supplementation to a *Z. bailii* diet supported animal development and promoted lipid accumulation despite the presence of the yeast (Fig 4J). Next, we assessed nutrient absorption in KTM-fed animals by supplementing the KTM lawn with the vital dye C1-BODIPY-C12, which is consumed with the food and can readily cross the intestinal apical membrane. Following a three-hour pulse of BODIPY, animals consuming *E. coli* OP50, KTM, and KTM-M all have detectable levels of BODIPY in their intestinal epithelial cells (Figs 4K and S7). Since animals consuming KTMs have very few, small lipid droplets the BODIPY staining was diffusely distributed throughout the intestinal cells; however, in the *E. coli* OP50 and KTM-M-fed animals the dye localized to intestinal lipid droplets and lysosome-related organelles [53,67].

Together, these findings are consistent with our previous observations that animals consuming KT microbes, either individually or in combination, are not impaired in their ability to absorb nutrients; but rather, the KTM diet likely restricts lipid accumulation by modulating host metabolic pathways.

## An intestinally driven metabolic response to KTM consumption

KTM-fed animals undergo normal development and show no detectable impairment in nutrient absorption, yet store markedly less lipids than control animals, including those fed the KTM-M diet. While our transcriptomics suggested that the expression of genes involved in development or reproduction were consistent across diets, we hypothesized that the expression of metabolic genes may be specifically altered by KTM consumption. Therefore, we performed additional analyses of our mRNA-Seq data derived from day one adult animals consuming either KTM, KTM-M, *A. tropicalis*, or the two *E. coli* diets to investigate if specific metabolic programs are altered by these diets. A PCA analysis revealed that the transcriptomes of animals fed the same diet cluster, with the transcriptomes of animals fed KTM, KTM-M, and *A. tropicalis* distinctly clustering apart from the transcriptomes of the *E. coli*-fed animals (Fig 5A), indicating that there is at least some commonality between the transcriptional responses of animals consuming any of the KT-associated diets that is different from *E. coli* diets. To eliminate the possibility of transgenerational epigenetic effects of the KTM diet, we compared the transcriptomes of animals fed KTMs for one generation to animals subjected to five generations of the KTM diet, finding no significant difference between these transcriptomes (Figs 5A and S8A).

Deeper investigation of our mRNA-Seq data revealed that each KT-associated diet did indeed result in some level of differential gene expression compared to the *E. coli* OP50 diet (*A. tropicalis*, 3,952 genes; KTM, 1,237 genes; KTM-M, 1,007 genes; 1% FDR; Figs 5B and S8B–S8F). Intriguingly, 295 genes were unique to the KTM diet (Fig 5B). Altered expression of these KTM-unique genes could be a major driver of the reduced lipid levels that we observed specifically in the KTM-fed animals. A gene ontology (GO) enrichment analysis [68] of the KTM-unique genes revealed an enrichment for genes annotated to have functional roles in lipid metabolism (Fig 5C). Since misexpression of core metabolic genes alters longevity and stress resistance pathways [69,70], we queried whether these same genes were also misexpressed in animals with reduced levels of DAF-2 (*i.e.*, the insulin receptor), which results in increased stress resistance, improved healthspan, and extended lifespan [71,72]. Indeed, depletion of DAF-2 in different tissues [72], including the intestine, results in transcriptional changes that are consistent with those seen in KTM-fed animals (S8G–S8I Fig). Together, these data suggest that consumption of fermenting KT microbes may remodel host lipid metabolism and stress resilience pathways to restrict fat accumulation and improve healthspan.

In *C. elegans*, the intestine functions as the primary hub for nutrient absorption, lipid storage, and metabolic regulation [52]. Our transcriptome data indicated that genes involved in lipid metabolism are modulated by KTM consumption, prompting us to investigate whether the host transcriptional response to KTMs occurs in the intestine. Using previously established gene expression data for the major tissues, we queried whether each set of diet-induced differentially expressed genes were enriched for a specific tissue [73,74]. We found that in response to KTM consumption there was a striking enrichment for differential expression of intestinal genes, as well as a depletion of neuronal and germline genes (Fig 5D). These data indicate that while genes expressed in the intestine are commonly differentially expressed in animals consuming KTMs, genes expressed in other tissue types tend not to be differentially expressed in KTM-fed animals.

To identify candidate genes that may be responsible for the metabolic effects of KTM consumption, we analyzed the expression levels of 5,676 genes that are annotated to function in

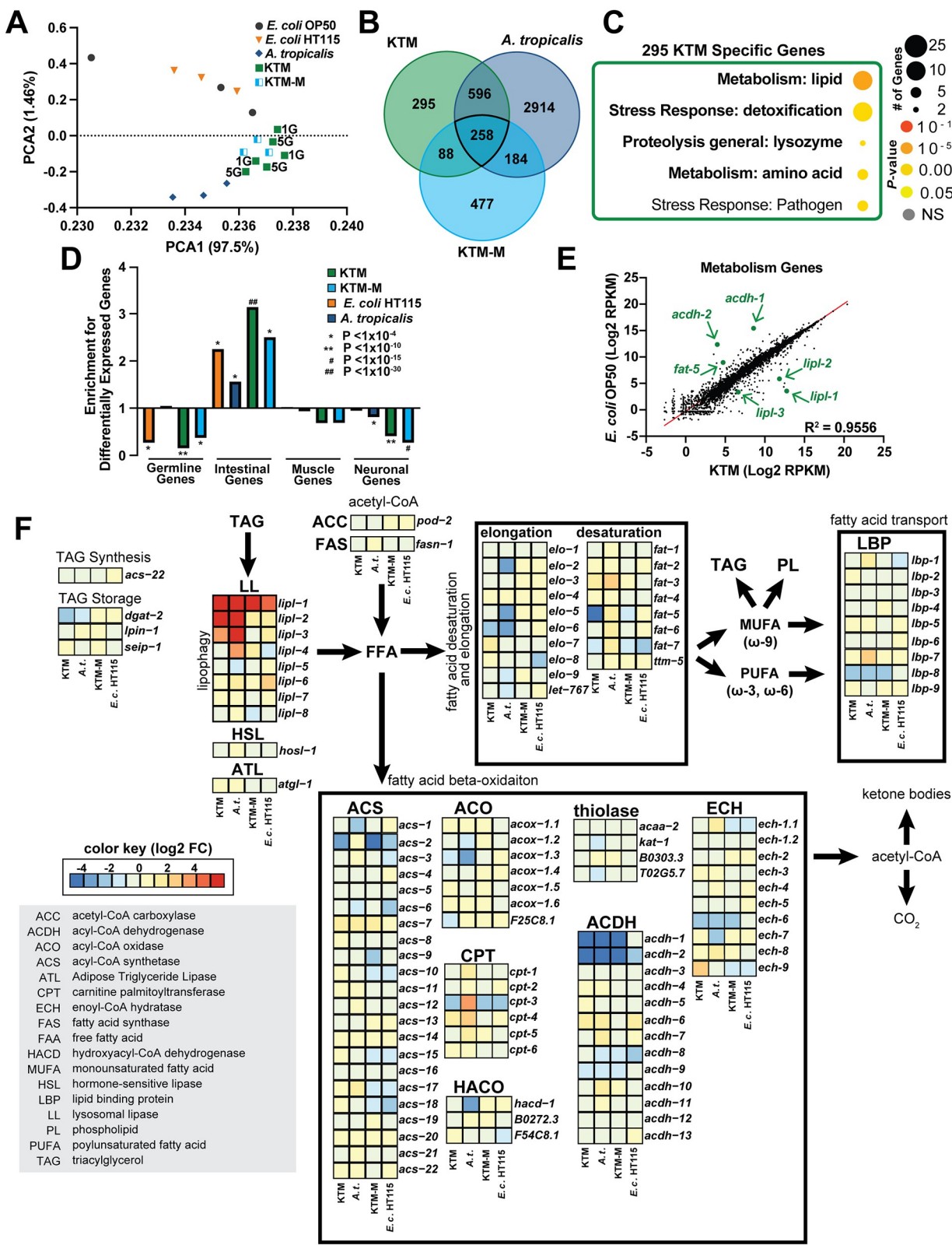

**Fig 5. Host lipid metabolism gene expression is modulated by KTM consumption.** (A) A Principal Component Analysis of the normalized mRNA-Seq data for the indicated diets (1G, KTM feeding for one generation; 5G, KTM feeding for five consecutive generations prior to collection).

(B) The overlap of the differentially expressed genes, determined relative to *E. coli* OP50, between each food source. (C) A Gene Ontology enrichment analysis performed on the 295 genes that are uniquely differentially expressed in animals consuming KTMs. (D) Enrichment for differential expression of genes that are expressed in the indicated tissues (observed/expected, hypergeometric *P* values reported). Values <1 indicate that genes expressed in the indicated tissue type tend not to be differentially expressed (under-enriched), while values >1 indicate tissues where differential expression is more common than expected by random chance (over-enriched). (E) A scatter plot and linear regression ($R^2$ = 0.9556) of the RPMK values for 5,676 metabolism-related genes (the genes of interest are indicated with arrows). (F) A schematic and gene expression heatmap (Log2 fold change values relative to *E. coli* OP50) for the indicated lipid metabolism genes for each diet (boxes from left to right: KTM, *A. tropicalis*, KTM-M, *E. coli* HT115). Raw data underlying panels A-F can be found in S5 Data and S4 Table.

metabolism [29]. This revealed that several genes known to function in lipid biology have altered expression in KTM-fed animals (Fig 5E and 5F). These included down-regulated genes that act in the β-oxidation of lipids (*acdh-1*, *acdh-2*), fatty acid desaturation (*fat-5*, *fat-6*, *fat-7*), or triglyceride synthesis (*dgat-2*), as well as up-regulated genes that act in lipolysis (*lipl-1*, *lipl-2*, *lipl-3*). These data suggest that expression of specific lipid metabolism genes in the intestine is modulated by KTM consumption. Consistently, intestinal expression of a GFP-based transcriptional reporter for the *acdh-1* gene, which encodes a conserved acyl-CoA dehydrogenase that catabolizes short chain fatty acids and branch chained amino acids, was reduced when animals were fed a KTM diet (S8J and S8K Fig). Together, our results suggest that transcriptional regulation of metabolic genes may, at least in part, underlie the reduction in intestinal lipids that we observed in KTM-fed animals.

## KTM consumption restricts lipid accumulation by regulating lipid droplet dynamics

Coordination of intestinal lipid stores is governed by both transcriptional and post-translational mechanisms that dynamically alter lipid droplets in response to external signals. Expansion of LDs is carried out via *de novo* lipogenesis and the action of acyl CoA:diacylglycerol acyltransferase (DGAT) enzymes, which catalyze the final step in TAG synthesis [75,76]. In contrast, lipases and lipophagy, a selective LD autophagy pathway, restrict LD size and number, respectively, and promote lipid catabolism [77–81]. Given that KTM-fed animals display a reduction in lipid levels and lipid droplet size, we reasoned that the expression of triglyceride lipases may be induced in response to KTMs; however, we found that expression of the adipocyte triglyceride lipase gene (*atgl-1/ATGL*), which encodes a LD-associated and starvation-responsive TAG lipase [52,82], and the hormone-sensitive lipase gene (*hosl-1/HSL*), which encodes a hormone-responsive TAG lipase, are not altered by KTM feeding (Fig 6A and 6B). We then inspected the expression of the remaining lipase genes within our mRNA-Seq data (Fig 5F), finding that three ATGL-like lipase genes (*i.e.*, *lipl-1*, *lipl-2*, and *lipl-3*) were markedly up-regulated in KTM-fed animals relative to those consuming the *E. coli* or KTM-M diets (Fig 6C–6E). Interestingly, *lipl-1,2,3* gene expression is known to increase upon fasting and the encoded proteins all localize to the lysosomes in the intestine where they break down LD-associated TAGs via lipophagy [83]. Consistent with these observations, expression of a single-copy P*lipl-1*::*mCherry* transcriptional reporter was specifically induced in the intestine in response to KTMs compared to the other food sources (Figs 6F and S9A). Up-regulation of the *lipl-1,2,3* lysosomal lipase genes, as well as the concomitant reduction in TAGs, suggests that KTM-fed animals may experience a fasting-like state even in the presence of sufficient nutrient availability.

To assess whether lysosomal lipases are required for the host response to KT, we used Oil Red O staining to determine the levels of intestinal lipids in previously generated *lipl* mutants [83,84]. We found that lipid levels were elevated in *lipl-1(tm1954); lipl-2(ttTi14801)* double mutants and *lipl-1(tm1954); lipl-2(ttTi14801); lipl-3(tm4498)* triple mutants relative to wild-type animals upon KTM consumption (Fig 6G). We also generated putative loss-of-function

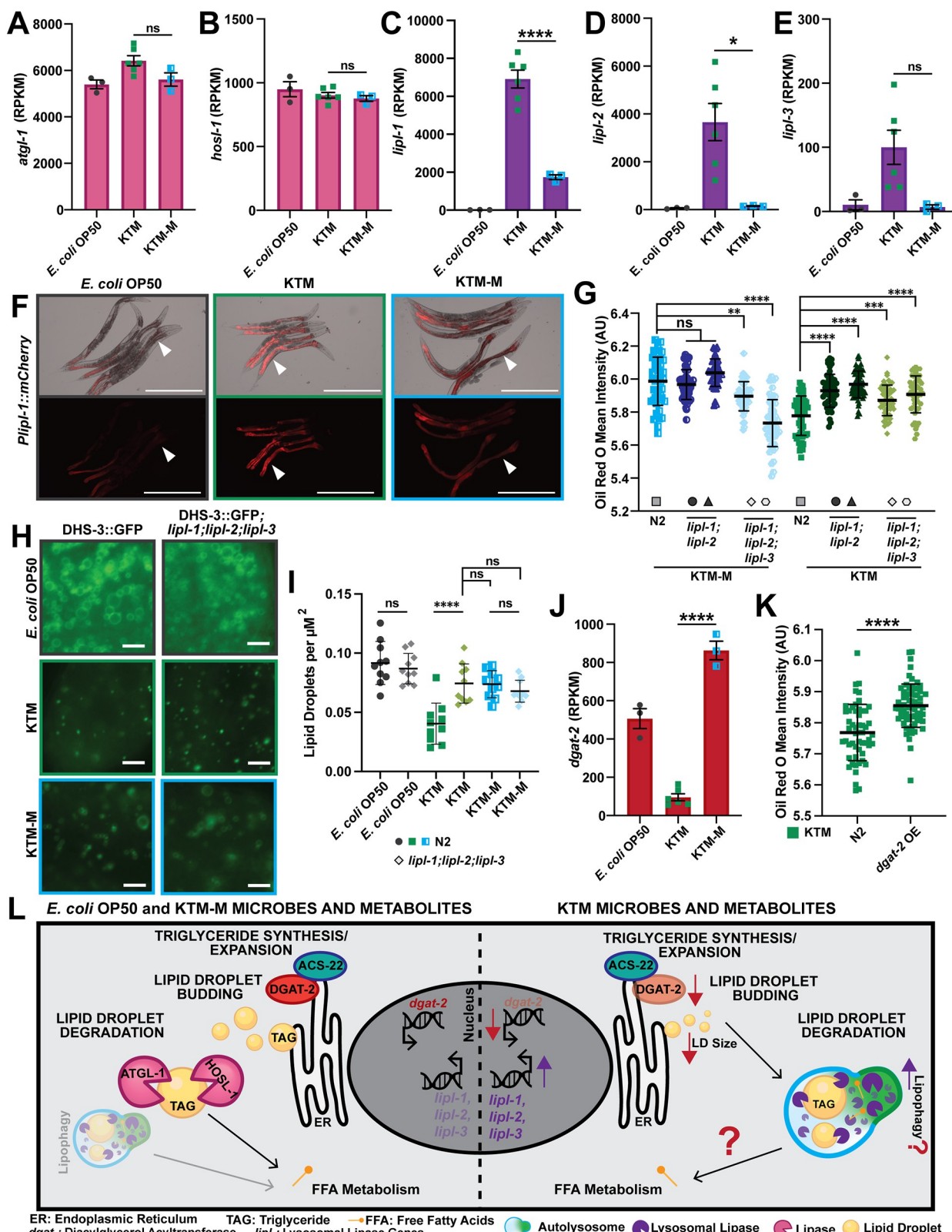

**Fig 6. KTM consumption stimulates host lipid catabolism and impairs TAG synthesis.** (A-E) Normalized gene expression values (RPKM, reads per kilobase of transcript per million mapped reads; mean ± SEM, ****, $P<0.0001$, *, $P<0.05$, ns, not significant, one-way ANOVA) for the

indicated lipase genes. (F) Representative images of animals expressing a P*lipl-1::mCherry* transcriptional reporter upon consumption of the indicated diets (white arrow heads point to the intestine; scale bars, 500μm). (G) Quantification of Oil Red O stained intestinal lipids in day 1 adult wild-type N2 and *lipl* mutant animals after consumption of the KTM-M (left group) or KTM (right group) diets (mean ± SD, ****, $P<0.0001$, ***, $P<0.001$, **, $P<0.01$, ns, not significant, one-way ANOVA). Data are shown for the following mutants: *lipl-1(tm1954) lipl-2(ttTi14801)* in circles, *lipl-1(rhd279) lipl-2(rhd282)* in triangles, *lipl-1(tm1954) lipl-2(ttTi14801) lipl-3(tm4498)* in diamonds, and *lipl-1(rhd279) lipl-2(rhd282) lipl-3 (tm4498)* in hexagons. (H) Representative images (scale bars, 5 μm) and (I) lipid droplet measurements (mean ± SD, ****, $P<0.0001$, ns, not significant, one-way ANOVA) of DHS-3::GFP-containing lipid droplets in wild-type N2 and *lipl-1(tm1954) lipl-2(ttTi14801) lipl-3(tm4498)* mutant animals. (J) Normalized gene expression values for the TAG synthesis gene *dgat-2* (mean ± SEM, ****, $P<0.0001$, one-way ANOVA). (K) Quantification of Oil Red O staining of intestinal lipids in wild-type N2 and DGAT-2::GFP transgenic animals, which constitutively overexpress DGAT-2 in the intestine (*dgat-2* OE; mean ± SD, ****, $P<0.0001$, T-test). (L) A model of KTM modulation of host lipid metabolism pathways showing 1) the induction of the lysosomal lipases that are essential to lipophagy and 2) the down-regulation of the TAG synthesis gene *dgat-2* thereby restricting lipid droplet initiation/expansion. Raw data underlying panels A-E, G, and I-K can be found in S6 Data.

nonsense mutations in the *lipl-1* and *lipl-2* genes using CRISPR/Cas-9, crossed these alleles to the existing *lipl-3(tm4498)* mutant [83], and performed Oil Red O staining of the resulting triple mutant. Consistent with our initial observations, simultaneous loss of *lipl-1,2* or *lipl-1,2,3* increased lipid levels in KTM-fed animals (Fig 6G). Since the LIPL-1,2,3 proteins localize to lysosomes and catabolize LD-associated TAGs, we reasoned that LD size or abundance may be altered in *lipl-1,2,3* mutants consuming KTMs. Therefore, we crossed the DHS-3::GFP reporter into the *lipl-1(tm1954); lipl-2(ttTi14801); lipl-3(tm4498)* triple mutant and measured intestinal LDs. Triple mutant animals fed the KTM diet, but not the KTM-M diet, had more LDs compared to wild-type animals; however, the LD size was similar between wild-type and mutant animals (Figs 6H–6I and S9B), suggesting that the LIPL-1,2,3 proteins promote LD degradation, but not LD shrinking, in KTM-fed animals. Together, these results indicate that up-regulation of the *lipl-1,2,3* lysosomal lipases in response to Kombucha Tea consumption partially governs the host metabolic response to KTMs and facilitates lipid catabolism.

In addition to induction of lipid catabolism pathways, Kombucha-associated microbes may impair TAG accumulation or LD expansion. To investigate this further using our mRNA-Seq data, we compared the expression levels of genes that are known to function in LD synthesis or expansion for animals fed *E. coli* OP50, KTMs, or the KTM-M [75,85]. Although levels of seipin (*seip-1*), lipin (*lpin-1*), and *acs-22/FATP4* (a long-chain fatty acid transporter and acyl-CoA synthetase enzyme) were not altered in response to KTM feeding, the *dgat-2/DGAT2* gene was dramatically and specifically down-regulated upon KTM consumption (Figs 6J and S9C–S9E), suggesting that TAG synthesis may be impaired in these animals. To test whether down-regulation of *dgat-2* restricts lipid accumulation in KTM-fed animals, we employed a strain that expresses *dgat-2* under the control of a constitutive intestinal promoter (P*vha-6::GFP::dgat-2*), which is not predicted to respond to KTM consumption. Indeed, constitutive expression of *dgat-2* partially suppressed the KTM-dependent depletion of intestinal lipid stores (Fig 6K). Together, these results support a model where the concomitant down-regulation of *dgat-2* and up-regulation of the lysosomal lipase genes limits TAG synthesis while promoting LD breakdown, which together restricts intestinal lipid accumulation in response to Kombucha consumption (Fig 6L).

## Discussion

The first records of Kombucha Tea consumption can be traced to ancient China where it was incorporated into common medical practices [86]. While its popularity has expanded throughout history, a recent surge in worldwide consumption makes it one of the most popular probiotic-containing fermented beverages, with its numerous purported human health benefits being a major contributor to its popularity [86]. Despite this long history and widespread anecdotal evidence that it improves metabolic health [9–13], little is known about whether Kombucha Tea consumption alters host metabolism and, if so, by which mechanisms this may occur.

To investigate Kombucha Tea's action in an animal model system, we established a reproducible method to deliver a diet of KT-associated microbes (KTM) to *C. elegans* though standard *ad libitum* feeding practices. Delivery of KTMs by feeding supports normal *C. elegans* development and fecundity, and importantly, results in robust KTM colonization of the intestinal lumen. Our study is the first to leverage a well-established animal model system to elucidate the molecular mechanisms of Kombucha Tea action in the host.

Here, we demonstrate that animals consuming KTMs are markedly devoid of lipids relative to animals fed other microbial diets, as determined by Oil Red O and Nile Red staining, biochemical triglyceride measurements, and size calculations of intestinal lipid droplets. Together, our results suggest that KTM consumption stimulates a fasting-like state in *C. elegans* that is distinct from traditional models of caloric restriction. Indeed, there are several lines of evidence that argue that KTM-fed animals are not experiencing caloric restriction, including 1) KTM feeding supports an increased rate of development for both wild-type and calorically restricted animals (*i.e.*, *eat-2* mutants), 2) KTM-fed animals are fertile (*i.e.*, they exhibit nearly normal brood sizes, reproductive lifespans, and expression of reproduction genes), 3) the individual KT microbes (*A. tropicalis*, *K. rhaeticus*, and *Z. bailli* supplemented with dead *E. coli*), as well as a simple mixture of the three microbes (KTM-M), fail to deplete host lipid stores, and 4) supplementation of KTMs with additional nutrients, either dead *E. coli* or higher concentrations of KTMs, did not increase lipid accumulation. Importantly, calorically restricted animals have severe growth and fertility defects [28,56–59], phenotypes that are inconsistent with those produced by KTM consumption. Finally, we found that host lipid utilization was maintained after washing the concentrated KT microbes with naïve, sucrose-only media prior to plating, supporting the hypothesis that the bioactive molecule(s) responsible for altering host lipid metabolism are intrinsic to the KTM microbes rather than found in the cell-free, fermented tea supernatant. Identification of these KTM-derived metabolites will be crucial to gain insight into the molecular mechanisms of KT action.

To gain a comprehensive view of the host metabolic response to Kombucha, we performed mRNA sequencing of animals consuming KTMs. While expression of developmental or reproduction genes were globally unchanged, expression of numerous lipid metabolism genes were specifically altered in response to KTMs, with a strong enrichment for genes known to function in the intestine. These include gene products that function in various aspects of lipid biology, including β-oxidation of lipids (*acdh-1* and *acdh-2*), fatty acid desaturation (*fat-5* and *fat-7*), triglyceride synthesis (*dgat-2*), and lipolysis (*lipl-1*, *lipl-2*, and *lipl-3*). The stearoyl-CoA desaturase genes, in particular *fat-5* and *fat-7*, were down-regulated in KTM-fed animals. This finding is notable since the *C. elegans* desaturases have lipid substrate preferences, and thus, differential expression of individual *fat* genes can result in alterations in the abundance of specific monounsaturated or polyunsaturated fatty acids [87]. FAT-5, which desaturates palmitic acid (16:0) to generate palmitoleic acid (16:1n-7), is transcriptionally down-regulated in KTM-fed animals, possibly resulting in a decrease in palmitoleic acid and increase in palmitic acid or other unsaturated fatty acids that are derived from palmitic acid. Specific changes in the abundance of monounsaturated or polyunsaturated fatty acids may contribute to the fasting-like state displayed by KTM-fed animals; however, lipidomic studies, paired with fatty acid supplementation experiments and genetic analyses, will be needed to resolve the role of the *C. elegans* desaturases in mediating the host response to KTM consumption.

In this study, we focused on three intestinal ATGL-like lipase genes *lipl-1*, *lipl-2*, and *lipl-3* that were specifically upregulated in KTM-fed animals, while the other 5 *lipl* genes, as well as the lipid droplet lipase genes *atgl-1* and *hosl-1*, remained unchanged. These findings argue that Kombucha consumption triggers a specific catabolic response to restrict lipid accumulation. The *lipl-1,2,3* genes encode three, likely redundantly acting lysosomal lipases that function in lipophagy-mediated break down of LD-associated TAGs [83]. Here, we demonstrate that the

*lipl-1,2,3* genes are partially required for KTM-mediated lipid catabolism, suggesting that lipophagy is induced by KTM consumption. Lipophagy, which is a selective form of autophagy that targets lipid droplet TAGs to liberate free fatty acids for further catabolism, is essential for lipid homeostasis and survival in times of low nutrient availability or during states of fasting. In addition to these conditions, homeostatic pathways can dynamically govern lipophagy induction under different nutrient- and stress-related conditions (*i.e.*, fed, fasted, and oxidative stress states) [80]. For example, *lipl-3* transcription can be governed by the interplay between the DAF-16/FOXO, PHA-4/FoxA, and HLH-30/TFEB transcription factors in specific contexts [80]. We propose that KTM consumption stimulates a fasting-like state in *C. elegans* to promote lipid utilization via lipophagy; however, future studies will be needed to dissect the precise molecular mechanisms that lead to lipophagy induction in response to KTMs. It's notable that a recent study by Xu and colleagues [19] in rodents lends substantial physiological evidence supporting the health claims made regarding human KT consumption, including protection against obesity and Type 2 Diabetes, which are disease states that are commonly associated with impaired lipid utilization or dyslipidemia [81,88–90]. Our discovery that *C. elegans* animals consuming a KTM diet may have elevated levels of lipophagy, and potentially a broader autophagy-driven metabolic reprograming, is consistent with these claims and suggests that future studies deconvoluting the host response to Kombucha consumption at the molecular level will provide insight into how Kombucha Tea may alter human metabolism.

Our mRNA-Seq data also revealed that *dgat-2*, which encodes an acyl-CoA:diacylglycerol acyltransferase (DGAT) enzyme, is dramatically down-regulated in response to KTM consumption. The DGAT enzymes catalyze the synthesis of triglycerides from diacylglycerol and a fatty acyl-CoA, resulting in TAG production and the expansion of lipid droplets. Constitutive over-expression of *dgat-2* increased lipid accumulation in animals consuming KTMs, suggesting that down-regulation of *dgat-2* expression, and consequently reduced TAG synthesis, may be part of the programmed host response to KT. Notably, induction of *dgat-2* in *C. elegans* supports the expansion of LDs in response to the pathogen *Stenotrophomonas maltophilia* [91], suggesting that *dgat-2* expression may be dynamically regulated by nutrient sensing or innate immunity pathways to govern lipid storage levels within LDs. It is possible that *dgat-2* expression is controlled by the same signaling networks that control the expression of the LIPL-1,2,3 lysosomal lipases, which together restrict the accumulation of lipids during KTM feeding. This could also explain why loss of *lipl-1,2,3* increases LD abundance but not size, as *dgat-2* likely remains down-regulated in these animals following KTM consumption.

Recently, it has become increasingly evident that *C. elegans* offers a powerful system to investigate potential human probiotic microbes to gain insight into their mechanisms of action and to identify potential human health benefits [23,25,26,31,92–94]. Our study establishes a rigorous, reproducible, and widely applicable system that leverages the genetic tractability of *C. elegans* to interrogate the physiological and mechanistic host response to probiotic microbes. While this is an exciting proposition, it is imperative to note that this work, as with other studies conducted using *C. elegans* as a model to investigate human-probiotic interactions, is not directly translatable to human health outcomes and offers no clinical advice or context for human Kombucha Tea consumption. We also acknowledge that the origin of this now popular fermented beverage has deep roots in ancient Chinese medical practices and was created by a culture different from our own. Therefore, we want to make it explicitly clear that we are not making judgements, conclusions or claims regarding Kombucha Tea's use in any human medical practices or its recreational consumption. Our findings do, however, offer exciting insights into possible mechanisms of KT microbe-mediated host metabolic reprograming and lays the foundation for future studies in mammalian model systems that could deconvolute the biological underpinnings of Kombucha Tea's potential health benefits.

## Materials and methods

### *C. elegans* strains and maintenance

All *Caenorhabditis* strains were maintained at 20°C on Nematode Growth Media (NGM) agar plates containing *E. coli* OP50 as previously described [95]. A full list of the strains used in this study are shown in S3 Table. All *C. elegans* strains were well-fed for at least three generations before use in experiments. Unless otherwise stated, eggs were harvested from gravid adults reared on *E. coli* OP50 by bleaching and animals were synchronized to the L1 stage by incubating the eggs overnight at room temperature. Preparation of L1 animals by bleaching was required to prevent *E. coli* contamination of Kombucha NGM plates in the following generation.

The P*lipl-1*::*mCherry* transgenic strain was constructed using Mos1-mediated Single Copy Insertion (mosSCI). The *lipl-1* promoter (1,228 bp; chromosome V: 12,918,779–12,920,006; WS288) was amplified by PCR and fused to *mCherry*::*unc-54 3'UTR* in pCFJ151 by Gibson Assembly. The resulting plasmid, pRD172[P*lipl-1*::*mCherry*::*unc-54 3'UTR* + *cb-unc-119(+)*], was microinjected into EG6699 to isolate the single-copy integrant *rhdSi53[Plipl-1*::*mCherry*::*unc-54 3'UTR* + *cb-unc-119(+)]* as previously described [96]. The *lipl-1(rhd279[A391*])* and *lipl-2(rhd282[A423*])* nonsense alleles were generated by CRISPR/Cas9 gene editing. Briefly, single-stranded oligonucleotide HR donor molecules and the Cas9::crRNA:tracrRNA complexes (crRNA sequence: 5'-UAGAGAACUUCUACUCAAAA-3') were microinjected into the germline of wild-type animals as previously described [97]. The HR donor sequence included a new XbaI cut site which allowed for genotyping via PCR followed by restriction digest prior to Sanger sequencing. The transcriptional reporter strains *rhdSi53[Plipl-1*::*mCherry*::*unc-54 3'UTR* + *cb-unc-119(+)]* and *wwIs24[Pacdh-1*::*GFP* + *cb-unc-119(+)]* were imaged with a Nikon SMZ-18 stereo microscope equipped with a DS-Qi2 monochrome camera at 10X zoom.

### Kombucha brewing

Kombucha was brewed using a serial fermentation method adapted from a homebrewing Kombucha kit (The Kombucha Shop). Ultrapure water (1L) was boiled for 3 minutes, removed from the heat, and dried tea leaves (2.5 g of Assam Black Tea and 2.5 g Green Tea) were steeped for 5 minutes using an infuser. After removal of the tea infuser, 128 g of granulated cane sugar (Domino) was dissolved in the tea and the solution was poured into a clean 5L glass brewing jar before 3L of chilled ultrapure water was added. Once the solution cooled to below 30°C, the SCOBY and ~500 mL of the previous fermented Kombucha broth was added to the brew jar and a tight weave muslin cloth was affixed to the jar opening to limit contamination during fermentation. The jar was then placed in indirect sunlight at room temperature (between 24–28°C) and allowed to ferment for a minimum of 8 days before a new culture was started, which allowed for complete fermentation and a pH of ~4.

### NGM Kombucha plates

For the single microbial diets, NGM plates were seeded with either *E. coli* strains (OP50 or HT115) after ~16 hours of growth at 25°C or with *A. tropicalis*, *K. rhaeticus*, or *Z. bailii* grown for at least 3 days at 25°C. The *E. coli* strains were grown in 25 mL of LB with shaking (250 rpm) while *A. tropicalis*, *K. rhaeticus*, and *Z. bailii* were grown in 25 mL of mannitol growth media (5 g Yeast Extract, 3 g Peptone, and 25 g Mannitol in 1 L) supplemented with 1% D-glucose and 1% glycerol with shaking (250 rpm). The strains were concentrated via centrifugation at 4,000 rcf for 5 minutes followed by resuspension in 5 mL of the appropriate media before

seeding on NGM plates. To calculate the microbial concentration of food sources, OD600 readings were taken followed by serial dilution and CFU quantification.

To prepare KTM NGM agar plates, 50 mL of the Kombucha Tea culture on day 2 or 3 of fermentation was removed and the microbes were concentrated via centrifugation for 5 minutes at 4,000 rcf. The supernatant was removed leaving 5 mL to resuspend the pelleted KTMs. Following resuspension via vortexing, 300 μL or 2 mL of concentrated KT was added to the middle of a 6cm or 10cm NMG plate, respectively. For 5x KTM plates, 250 mL of culture was concentrated to 5 mL. Plates were allowed to mature for 4 days at room temperature before being used in experiments. The KTM-M NGM plates were prepared by first individually growing up 20 mL cultures of *A. tropicalis*, *K. rhaeticus*, and *Z. bailii*. The microbes were then concentrated by centrifugation, resuspended with filter sterilized tea media (2.5 g of Assam Black Tea, 2.5 g Green Tea, 128 g of granulated cane sugar, 1L of water), combined into a single culture, washed with sterilized tea media, reconcentrated by centrifugation, resuspended in 5 mL of the supernatant, seeded onto NGM plates, and incubated for 4 days at room temperature. To confirm that the filter sterilized tea media was free from microbes and/or spores, the sterilized tea media was plated on NGM plates and monitored for growth over 14 days, which resulted in plates free of microbial growth.

Similarly, microbes were grown independently, harvested, and combined in sterilized tea media to generate the small-scale KTM-FM cultures, which were maintained in 50 mL conical tubes with loosely tapped lids at room temperature. At different timepoints, 30 mL was removed from the culture and replaced with 30 mL of fresh sterilized tea media. The following day 25 mL was removed from the culture, concentrated by centrifugation, seeded onto NGM plates, and incubated at room temperature for 4 days prior to use. A long-term, established KTM-FM culture was started in a similar fashion, but the culture was fermented in a 500 mL graduated cylinder covered in a cheese cloth, which was serially fermented over time by removing 50 mL of fermented culture (used for plates) prior to the addition of 50 mL of fresh sterilized tea media.

## 16S rDNA sequencing of Kombucha culture and plates

The Kombucha Tea culture was initiated and KTMs were seeded onto plates as described above. For the day 1 culture timepoint, 1 mL of 10x concentrated Kombucha was subjected to further centrifugation at 16,000 krcf for 10 minutes, the supernatant was removed, and the pellet was flash frozen in liquid nitrogen. The KTM plates were prepared for 16S sequencing at the same time using 10x concentrated Kombucha. For the subsequent culture sampling, 10 mL of KT was collected and the KTMs were harvested by centrifugation. For KTM plate samples, the microbes were removed from NGM plates at different timepoints using a cell scrapper and were collected into 1 mL of UltraPure DNase/RNase free water, concentrated by centrifugation, and frozen. All 16S rDNA sequencing was performed by the UNC Microbiome Core on an Illumina MiSeq instrument (PE 250). The data analysis was performed on 32,000–95,000 raw reads per sample using Qiime2 [98].

## Lawn avoidance assay

Approximately 50 synchronized L1 animals were dropped outside of each microbial lawn and the number of animals on each lawn was counted at 48, 72, and 96 hours later. The proportion of animals off the lawn was calculated as $N_{off\ lawn}/N_{total}$ for each timepoint. Each biological replicate was averaged from three technical replicates and the data were plotted as the mean ± SEM using Prism 9. An ordinary one-way ANOVA followed by Sidak's multiple comparisons test was used to calculate statistical significance between groups.

## Food choice assay

NGM plates were seeded in four quadrants, each with 30 μL of one of the four food sources (*E. coli* OP50, HT115, *A. tropicalis*, and KTMs). Approximately 50 synchronized L1 animals were dropped in the middle of the plate and the fraction of animals on the different microbial lawns was counted 48 hours later.

## Pumping rate measurements

Animals were grown on each diet from synchronized L1s and the pumping rate of 15 day one adult animals was manually counted using a Nikon SMZ800N Stereo microscope. The number of pharyngeal contractions over a one-minute span was counted and data were plotted as the mean ± SD using Prism 9. A one-way ANOVA followed by Tukey's multiple comparisons test was used to calculate statistical significance between groups.

## Gut colonization assay

Measurement of the bacterial loads in *C. elegans* animals after consumption of each diet was performed as previously described [99]. Briefly, ~150 animals were grown from synchronized L1s on each diet to adulthood and ~30 animals were picked to an empty plate for 30 minutes to minimize bacterial transfer from lawn. Ten animals were hand-picked to M9 media containing 100 μg/mL levamisole, allowed to settle, and were washed three times with M9 media containing levamisole and gentamicin (100 μg/mL). Animals were lysed in 250 μL 1% Triton X-100 using thirty 1.5 mm sterile zirconium oxide beads (Next Advance) with an electric benchtop homogenizer (BioSpec Mini-beadbeater). The 1.5 mL tubes were shaken twice for 90 sec before serial dilution of the lysates and plating onto standard NGM plates. The CFUs/animal values were calculated as described [99]. Data were plotted using Prism 9 and the statistical significance between food sources was determined by one-way ANOVA followed by Sidak's multiple comparisons test.

## SEM imaging of the *C. elegans* Intestine

Day 1 adult animals were fixed with 2% paraformaldehyde in 150 mM sodium phosphate buffer (PB, pH 7.4) at room temperature and stored at 4°C. Samples were washed 3 times with PB, followed by 3 water rinses, dehydrated using an ethanol gradient (30%, 50%, 75%, 100%, 100%, 100%), washed with two hexamethyldisilazane (HMDS) exchanges, and allowed to dry in HMDS. Dried animals were brushed onto double-sided carbon adhesive mounted to a 13 mm aluminum stub and a scalpel was used to slice the *C. elegans* animals open by drawing the blade upward though the body of the animal while they were adhered to the adhesive. Mounted samples were then sputter coated with 5 nm of gold-palladium (60 Au:40 Pd, Cressington Sputter Coater 208HR, model 8000–220, Ted Pella Inc). Images were taken using a Zeiss Supra 25 FESEM operating at 5 kV, using the InLens detector, ~7 mm working distance, and 30 μm aperture (Carl Zeiss SMT Inc) at 5,000X and 15,000X zoom.

## Calcofluor White Staining

Calcofluor White (or Fluorescent Brightener #28), which stains chitin and cellulose, was added to levamisole paralyzing solution at approximately 1 mg/mL. The animals fed different diets were picked to agar pads containing levamisole and Calcofluor White, covered with a coverslip, and stained for 10 minutes. For *K. rhaeticus* and *E. coli* OP50 imaging (S6J Fig), animals were imaged with a Leica DMI8 with an xLIGHT V3 confocal microscope with a spinning disk head (89 North) equipped with a Hamamatsu ORCAFusion GENIII sCMOS camera using a 63X oil objective (Plan-Apochromat, 1.4 NA). For imaging of the KTMs (S6L Fig),

animals were imaged with a Ti2 widefield microscope equipped with a Hamamatsu ORCA--Fusion BT camara using a 100X oil objective (Plan Apo λ). Importantly, the stain was prone to rapid photobleaching, and thus, areas of interest were found using the DIC channel and animals were only exposed to the florescent light during image acquisition. Nikon Elements was used to denoise and deconvolute the KTM images and both sets of images were processed in Fiji v2.9.0 [100] to introduce pseudo coloring.

### Oil Red O and Nile Red staining

Approximately 150 animals were grown from synchronized L1s on NGM plates containing different food sources for 72 hours at 20˚C. Day 1 adult animals were washed off the plates in M9 media, allowed to settle on ice, washed three times with S-basal media, and fixed in 60% isopropanol. For Oil Red O staining, fixed animals were treated with filtered 0.5% Oil Red O for 7 hours before washing the animals with 0.01% Triton X-100 in S-basal as previously described [101]. For Nile Red staining, isopropanol-fixed animals were stained for 2 hours with fresh Nile Red/isopropanol solution (150 μL Nile Red stock at 0.5 mg/mL per 1 mL of 40% isopropanol) [29]. For whole body analyses, animals were mounted on agar pads and imaged for Oil Red O staining at 3X zoom with a Nikon SMZ-18 Stereo microscope equipped with a DS-Qi2 monochrome camera or for Nile Red staining at 4X zoom using a Ti2 widefield microscope equipped with a Hamamatsu ORCA-Fusion BT camera. Color images of Oil Red O-stained animals were obtained at 10X magnification using the Ti2 widefield microscope equipped with a Nikon DS-FI3 color camara. For analysis of intestinal Oil Red O staining (Fig 6G and 6K), animals were imaged at 10x with the Ti2 widefield microscope equipped with a Hamamatsu ORCA-Fusion BT camera.

For quantification of Oil Red O staining, whole animals were outlined using Fiji and the average gray value (0 to 65,536) for each individual was measured. The resulting values were subtracted from 65,536, which inverts the scale so that strongly stained animals now have higher values. True background values are the unstained regions within each animal; however, these regions are impossible to identify objectively. Thus, no background subtraction was performed, which compresses the data to a small range of values (55,000 to 65,000, which we report 5.5 to 6.5). We found this approach to be highly reproducible. For the intestinal Oil Red O staining analyses (Fig 6G and 6K), the mean gray values were calculated in a box (25x25 pixels) drawn within the first two intestinal cells and the analysis was performed as described above. For Nile Red staining, average fluorescence intensities were also measured using Fiji and no background subtraction was performed. All data were plotted using Prism 9 as the mean ± SD and a one-way ANOVA followed by Tukey's multiple comparisons statistical test was performed for each experiment. For each set of staining experiments, at least three biological replicates were performed and yielded similar results.

### Quantification of Lipid Droplets

Live day 1 adult *ldrIs1[Pdhs-3::dhs-3::GFP]* animals were mounted on agar pads with levamisole and imaged with a Ti2 widefield microscope equipped with a Hamamatsu ORCA-Fusion BT camara using a 63X oil objective. Bright field DIC and GFP images capturing the last two intestinal cell pairs were imaged in 0.2 μm slices using the same settings across samples. Following acquisition, Nikon Elements was used to select a representative slice in the middle of the stack for downstream analysis using Fiji. The DIC image was used to outline the intestinal cell pair and the diameter of the lipid droplets were measured by hand in the GFP channel using the line tool and ROI manager. Measurements were collected from the last intestinal cell pair for 10 representative animals for each food source and the average lipid droplet

measurements for each animal was plotted using Prism 9. A one-way ANOVA followed by Tukey's multiple comparisons test was used to compare groups. Two independent experiments were conducted with similar results.

## Biochemical triglyceride measurements

Day 1 adult animals consuming each food source were harvested and processed as previously described [55]. A Triglyceride Assay Kit was used to measure triglycerides per the manufactures' instructions (Abcam, ab65336). Three biological replicates were performed, the data were plotted in Prism 9, and a one-way ANOVA followed by Tukey's multiple comparisons test was carried out to compare groups.

## Molting assays

Between 1–8 synchronized *mgIs49[Pmlt-10::GFP::PEST]* L1s were dropped into each well of a 24-well plate containing NGM media and seeded with 20 μL of each food source (one plate per food source). Animals were reared at 20°C and visualized by fluorescence microscopy every hour for 70 hours on a Nikon SMZ-18 Stereo microscope. At each time point, animals were scored as green (molting) or nongreen (not molting). Wells without animals were censored. The fraction of animals molting for each timepoint was calculated and plotted with Prism 9. The experiments were performed at least twice (except for KTM-M and KTM-FM) with similar results.

## Developmental timing measurements

Wild-type N2 and *eat-2(ad465)* mutants were grown to adulthood and egg prepped as described above. The caloric restriction plates containing $10^8$ or $10^9$ CFUs/mL of *E. coli* OP50 were prepared as previously described [59]. Approximately 20 synchronized L1s were dropped on each food source in technical triplicates and grown at 20°C. After exactly 48 hours the animals were scored based on vulva morphology as young adults, L4 larval stage, or less than L4 larval stage. The percent of animals at each larval stage was calculated for three biological replicates and the data were plotted as the mean ± SEM using Prism 9.

## Brood size measurements

Animals were grown on their respective food sources for 48 hours at which time 15 L4s were singled to the corresponding food source and allowed to mature. The animals were moved to fresh plates every 24 hours for 6 days. Two days after the adult hermaphrodite was moved to a new plate, the L3/L4 progeny were counted and removed. The unhatched eggs were not counted. Total progeny for each individual hermaphrodite was plotted as mean ± SD using Prism 9 and a one-way ANOVA test with a Tukey's multiple comparison correction was performed. The average reproductive output per day was also calculated and an unpaired T-test was performed to identify differences between these means.

## VIT-2::GFP quantification

Animals expressing VIT-2::GFP at endogenous levels (strain BCN9071) were grown to adulthood, egg prepped, and hatched over night at room temperature as described above. The starved L1s were dropped on NGM plates seeded with their respective food sources and grown for 72 hours at 20°C. Gravid day 1 adult animals were washed off the NGM plates and eggs were liberated by bleaching. Following three washes with M9 media, embryos were mounted on agar pads and imaged with a Ti2 widefield microscope equipped with a Hamamatsu ORCA-Fusion BT camara using a 20X objective. Bright field DIC and GFP images were

captured and the fluorescent intensity of 30 early-stage embryos (prior to the 44-cell stage) for each condition was measured using Fiji. The mean fluorescent intensity ± SD was plotted using Prism 10 and statistical differences between the groups was calculated using an unpaired T-test. Three independent biological replicates were performed and yielded similar results.

### Intestinal lumen measurements

Day 1 adult ERM-1::GFP animals (strain BOX213) consuming each diet were mounted on agar pads with levamisole and imaged using a Nikon SMZ-18 Stereo microscope equipped with a DS-Qi2 monochrome camera. Bight field and GFP images were acquired for at least 10 individuals. Three measurements of the intestinal lumen diameter (positioned at the anterior intestine, the vulva, and the posterior intestine) were performed using Fiji. For each individual, a ratio of the lumen width relative to body width was calculated at each of the three positions along the animal and the three values were averaged. Ten individuals were measured for each biological replicate and the data are reported as the mean ± SEM of three biological replicates. A one-way ANOVA test with a Tukey's multiple comparison correction was performed in Prism 10.

### BODIPY staining

C1-BODIPY-C12 (Thermo Fisher, D3823) was resuspended in DMSO to generate a 10 mM stock solution. This solution was diluted in S-basal media and overlaid onto the microbial lawn to produce a final concentration of 10 μM within the NGM plate. The plates were allowed to dry for at least 1 hour in the dark before day 1 adult animals were picked to the BODIPY plates. After 3 hours, animals were mounted on agar pads with levamisole and imaged on a Ti2 widefield microscope equipped with a Hamamatsu ORCA-Fusion BT camara using a 40X oil objective. Bright field DIC and GFP images were captured for at least 20 animals at both the anterior and posterior sections of the intestine. Representative images showing detectible levels of BODIPY staining were selected from two independent experiments and are displayed in Figs 4K and S7.

### Whole genome sequencing and analysis of Kombucha microbes

Mannitol growth media supplemented with 1% D-glucose and 1% glycerol was inoculated with the KT associated microbes and the cultures were grown for 48 hrs at 25˚C with shaking. The gDNA was isolated from cell pellets using the Wizard Genomic DNA Purification kit (Promega, A1120). Preparation and Illumina short read sequencing (PE 150) of DNA-Seq libraries was performed by Novogene (Sacramento, CA). Initially, an unbiased metagenomic analysis was performed using Kraken 2 [65] to identify candidate microbial species for each Kombucha-associated microbe. Next, we downloaded the whole genome reference sequences for various strains for each candidate species from the NCBI Genome database and mapped our reads against those reference genomes using Bowtie 2 with the default settings [102]. The overall alignment rate generated by the Bowtie 2 algorithm was reported. The whole genome sequencing data are available at the Sequencing Read Archive (PRJNA1044129).

### mRNA sequencing

Wild-type N2 animals were grown on 10 cm NGM agarose plates (1000 animals/plate) in the presence of their respective food sources. Day 1 adults were harvested, washed 3 times in M9 buffer, and flash frozen. The total RNA was isolated using Trizol (Thermo Fisher), followed by two rounds of chloroform extraction, RNA precipitation with isopropanol, and an 80% ethanol wash of the RNA pellet. In some cases, an RNA Clean & Concentrator-25 kit (Zymo,

R1017) was used to increase the purity of the sample. The mRNA-Seq libraries were prepared and sequenced by Novogene (Illumina, PE 150). The data were processed exactly as previously described [103]. RPKM values and identification of differentially expressed genes (1% FDR) were calculated using the DESeq2 algorithm [104], which can be found in S4 Table. Lists of developmental, reproduction, and metabolic genes have been previously described [29], and scatter plots showing expression levels of these genes were generated using the DESeq2 RPKM values. Heatmaps and PCA plots were generated with the pheatmap [105] and tidyverse [106] R packages, respectively. All other plots showing mRNA-Seq data were made in Prism 9. Raw and processed mRNA-Seq data have been deposited in GEO (GSE236037).

## Supporting information

**S1 Fig. The phylogenetic profile of the KTMs on NGM plates is similar to that of the KT culture.** (A) Images of NGM worm plates seeded with a KTM lawn. The preparation starts at day 0 when a new KT brew cycle is initiated, the microbes are seeded on day 1, and incubated at room temperature to day 5 before the KTM plates are used. (B) Representative photos of KT brews at day 1 and day 7 of fermentation. The KTMs are extracted from the culture at day 1 and plated. (C) A comprehensive view of 16S rDNA sequencing results of the KT microbes from fermenting Kombucha culture, seeded NGM plates, or the pellicular biofilm from the Kombucha culture. The plot shows the frequency of each species (8 most abundant microbes displayed; a complete list can be found in S1 Table). (D) A plot of Faith's phylogenetic diversity index showing the difference in α-diversity between the indicated samples (**, p<0.01, one-way ANOVA). (E) The Pielou Evenness Diversity Index, measuring the microbial diversity and species richness in the indicated samples (ns, not significant, one-way ANOVA). Raw data underlying panels C-E can be found in S7 Data. (TIF)

**S2 Fig. Worms choose other diets over a KTM diet.** (A) A schematic depicting the food choice assay. (B) The portion of wild-type N2 animals at the L4 stage on each food source 48 hours after dropping L1s (n>200/trial, 3 biological replicates). (C-E) Food choice assays for the N2, MY10, and JU1212 *C. elegans* strains scored at the L4 stage (48h post L1 drop, n>150/trial, 3 biological replicates). (F-I) The portion of L4 stage worms on each food source at 48h post L1 drop for the N2 *C. elegans*, PB2801 *C. brenneri*, AF16 *C. briggsae*, and PB4641 *C. remanei* strains (n>75/trial, 3 biological replicates). All food choice data are plotted as the mean ± SEM. All food choice assays include n>150 animals per replicate and the data are plotted as the mean ± SEM (****, *P*<0.0001, ***, *P*<0.001, **, *P*<0.01, *, *P*<0.05, ns, not significant; one-way ANOVA). Raw data underlying panels B-I can be found in S8 Data. (TIF)

**S3 Fig. Host lipid distributions during reproduction and across individuals.** (A) Quantification of day 3 and day 5 adults stained with Oil Red O (mean ± SD, ****, *P*<0.0001, one-way ANOVA). (B) Measurements of individual lipid droplet sizes measured across ten individuals consuming *E. coli* OP50, KTMs, or KTM-Mix (mean ± SD, n = 10 animals/trial, 2 biological replicates). The distribution of lipid droplet sizes is similar across individuals fed the same diet. Raw data underlying panels A and B can be found in S9 Data. (TIF)

**S4 Fig. Average progeny per day.** A table displaying the average progeny laid per day of the reproductive period demonstrates that KTM-fed animals exhibit a similar egg laying rate relative to *E. coli* OP50-fed animals (mean, ****, *P*<0.0001, ***, *P*<0.001, **, *P*<0. 01, *, *P*<0.05, ns, not significant, T-test). Raw data underlying the figure can be found in S10 Data. (TIF)

**S5 Fig. rDNA sequencing identifies candidate KTMs.** Results from 16S rDNA sequencing of the isolated bacterial KTMs indicate that (A) *A. tropicalis* and (C) a member of the *Komagataeibacter* genus are components of our Kombucha culture. (B) Sequencing of the ITS region of the KTM yeast isolate revealed that the strain belongs to the *Brettanomyces* or *Zygosaccharomyces* genus. Raw data underlying panels A-C can be found in S1 Table.
(TIF)

**S6 Fig. Deconvolution of Kombucha Tea facilitates the creation of fermenting and non-fermenting mixes of KTM.** (A-B) Measurements of the microbial concentrations in each of the indicated microbial mixes or single microbial cultures (mean ± SEM, ****, $P<0.0001$, one-way ANOVA). (C) Oil Red O staining of day 1 adult animals fed an *E. coli* OP50, KTM-Mix, or KTMs diet, as well as a 5X concentrated version of the KTM diet (mean ± SD, ****, $P<0.0001$, one-way ANOVA). Increasing the concentration of KTMs decreases lipid storage. (D) Representative images of animals off and on a lawn of *Z. bailii* yeast 72 hours post L1 drop, which shows that animals fail to develop when consuming a *Z. bailii* diet (worms are indicated with white arrow heads; scale bar, 500 μm). (E) Lipid droplet density measurements with each datapoint representing the number of lipid droplets per μm$^2$ for the last two intestinal cells of animals consuming a KTM or KTM-M diet (the KTM data are also shown in Fig 2H; mean ± SD, **, $P<0.01$, T-test). (F) A scatter plot comparing the expression of 2,229 developmental genes in animals fed *E. coli* versus KTM-M as determined by mRNA-Seq (RPKM, reads per kilobase of transcript per million mapped reads). A linear regression analysis and the corresponding $R^2$ value is reported. (G) A choice assay showing the portion of wild-type N2 animals at the L4 stage on the indicated food sources 48 hours after dropping L1s (n>200/trial, 3 biological replicates; mean ± SEM, ****, $P<0.0001$, *, $P<0.05$, ns, not significant, one-way ANOVA). (H-I) The developmental rate of animals expressing a P*mlt-10*::*GFP-PEST* reporter when fed a KTM, KTM-Mix, or a KTM-FM diet. Synchronized L1 worms were reared at 20°C for ~72 hours and scored hourly. (J) Representative images of animals consuming *K. rhaeticus* or *E. coli* OP50 after staining with Calcofluor White, which selectively labels intestinal microbes producing chitin or cellulose (white arrow heads indicate the intestinal lumen; scale bars, 10 μm). (K) Representative brightfield DIC images showing yeast cells in the intestine of animals consuming KTMs and yeast cells on the slide (gray arrow heads indicate yeast cells, magnified inset image shown for clarity, scale bar 5 μm). (L) Representative images of animals consuming KTMs after staining with Calcofluor White (gray arrow heads in the inset indicate yeast cells; scale bars, 5 μm). (M) Intestinal lumen width measurements of animals consuming the *E. coli* OP50, KTM, and KTM-M diets. Data are reported as the percent of the body width taken up by the intestinal lumen (mean± SEM, **, $P<0.01$, ns, not significant, one-way ANOVA). Raw data underlying panels A-C, E-I, and M can be found in S11 Data.
(TIF)

**S7 Fig. BODIPY lipids are absorbed into intestinal cells of KTM-fed animals.** Representative DIC and fluorescence images showing C1-BODIPY-C12 absorption into the intestinal epithelial cells of animals feeding on an *E. coli* OP50, KTM, or KTM-M diet. The pink stars indicate BODIPY remaining in the intestinal lumen, the pink arrowheads point to partial BODIPY absorption into the intestinal cells, white stars indicate a lack of BODIPY remaining in the intestinal lumen, and white arrowheads point to fully stained cells that have absorbed BODIPY (scale bars, 10 μm).
(TIF)

**S8 Fig. KTM consumption results in widespread changes in gene expression.** (A) A scatter plot and linear regression analysis comparing the expression of all genes in animals fed KTMs for one

generation (1G) or for five generations (5G), suggesting that pervasive transgenerational epigenetic regulation of gene expression by KTMs is unlikely. (B-F) Volcano plots showing the differentially expressed genes for the indicated samples relative to the *E. coli* OP50 sample. (G) Enrichment (observed/expected, hypergeometric *P* values reported) for differentially expressed genes common between KTM-fed animals and animals depleted of DAF-2::AID in the indicated tissues using the auxin degron system [72]. Values >1 indicate over-enrichment, or that the same genes tend to be differently expressed in both animals consuming KTMs and animals depleted of DAF-2 compared to random chance. The overlap between differentially expressed genes that are either (H) up-regulated or (I) down-regulated in animals consuming KTMs and animals depleted DAF-2::AID in the intestine (hypergeometric *P* values are shown). (J) Representative fluorescent images (scale bar, 500 μm) and (K) quantification of the acyl-CoA dehydrogenase P*acdh-1*::*GFP* reporter on the indicated microbial diets (n = 40, mean ± SD, ****, *P*<0.0001, ns, not significant, one-way ANOVA). Raw data underlying panels A-I and K can be found in S12 Data.
(TIF)

**S9 Fig. Expression of the *lipl-1* gene is modulated in the intestine upon KTM consumption, but the lysosomal lipases genes *lipl-1,2,3* are not required to restrict lipid droplet size.** (A) Quantification of the expression levels of the lysosomal lipase P*lipl-1*::*mCherry* reporter in animals grown on *E. coli* OP50, KTM, and KTM-M (n>200, mean ± SD, ****, *P*<0.0001, one-way ANOVA). (B) Lipid droplet size measurements in wild-type N2 and *lipl-1(tm1954) lipl-2 (ttTi14801) lipl-3(tm4498)* mutant animals with each datapoint representing the average intestinal lipid droplet diameter for a single animal (mean ± SD, ***, *P*<0.001, ns, not significant, one-way ANOVA). (C-E) Normalized gene expression values for the indicated TAG synthesis genes (mean ± SEM, ***, *P*<0.001, *, *P*<0.05, ns, not significant, one-way ANOVA). Raw data underlying panels A-E can be found in S13 Data.
(TIF)

**S1 Table. An Excel spreadsheet containing the taxonomy report from 16S rDNA sequencing.** Shown are individual sequencing results for biological replicates of the Kombucha Tea cultures, the Kombucha Tea biofilm (one replicate), and Kombucha Tea-associated microbes isolated from *C. elegans* NGM plates.
(XLSX)

**S2 Table. Sequencing read alignment rates from whole genome sequencing of Kombucha Tea-associated microbes.**
(PDF)

**S3 Table. The *C. elegans* strains used in this study.** The strain names, genotypes, and associated references are shown.
(PDF)

**S4 Table. The DESeq2 outputs from the mRNA-Seq analysis.** An Excel spreadsheet containing, in separate tabs, gene counts (RPKM, reads per kilobase of transcript per million mapped reads) for all genes, as well as the differential gene expression calls for the following comparisons: *E. coli* OP50 vs. *E. coli* HT115, *E. coli* OP50 vs. *Acetobacter tropicalis*, *E. coli* OP50 vs. KTM, and *E. coli* OP50 vs. KTM-M.
(XLSX)

**S1 Data. Excel spreadsheet containing, in separate tabs, the numerical data underlying Fig 1D, 1E and 1F.**
(XLSX)

**S2 Data. Excel spreadsheet containing, in separate tabs, the numerical data underlying Fig 2B, 2D, 2E, 2G and 2H.**
(XLSX)

**S3 Data. Excel spreadsheet containing, in separate tabs, the numerical data underlying Fig 3A, 3B, 3C, 3D, 3E, 3F, 3G, 3H, 3I, 3J, 3K, 3L, 3M and 3N.**
(XLSX)

**S4 Data. Excel spreadsheet containing, in separate tabs, the numerical data underlying Fig 4C, 4E, 4F, 4G, 4I and 4J.**
(XLSX)

**S5 Data. Excel spreadsheet containing, in separate tabs, the numerical data underlying Fig 5A, 5B, 5C, 5D, 5E and 5F.**
(XLSX)

**S6 Data. Excel spreadsheet containing, in separate tabs, the numerical data underlying Fig 6A, 6B, 6C, 6D, 6E, 6G, 6I, 6J and 6K.**
(XLSX)

**S7 Data. Excel spreadsheet containing, in separate tabs, the numerical data underlying S1C, S1D and S1E Fig.**
(XLSX)

**S8 Data. Excel spreadsheet containing, in separate tabs, the numerical data underlying S2B, S2C, S2D, S2E, S2F, S2G, S2H and S2I Fig.**
(XLSX)

**S9 Data. Excel spreadsheet containing, in separate tabs, the numerical data underlying S3A and S3B Fig.**
(XLSX)

**S10 Data. Excel spreadsheet containing, in separate tabs, the numerical data underlying S4 Fig.**
(XLSX)

**S11 Data. Excel spreadsheet containing, in separate tabs, the numerical data underlying S6A, S6B, S6C, S6E, S6F, S6G, S6H, S6I and S6M Fig.**
(XLSX)

**S12 Data. Excel spreadsheet containing, in separate tabs, the numerical data underlying S8A, S8B, S8C, S8D, S8E, S8F, S8G, S8H, S8I and S8K Fig.**
(XLSX)

**S13 Data. Excel spreadsheet containing, in separate tabs, the numerical data underlying S9A, S9B, S9C, S9D and S9E Fig.**
(XLSX)

## Acknowledgments

Some of the strains used in this study were provided by the *Caenorhabditis* Genetics Center, which is supported by the NIH Office of Research Infrastructure Programs (P40 OD010440). The *lipl-3(tm4498) lipl-2(ttTi14801) lipl-1(tm1954)* strain was generously provided by Dr. Eyleen O'Rourke (UVA). We want to thank Dr. Anne Matthysse (UNC) for her helpful suggestions regarding the isolation of KT-associated microbes and for some of the reagents used

in this study, including the Calcofluor White stain and driselase. The authors would also like to thank Kristen K. White and the Microscopy Services Laboratory for their assistance with SEM preparation and imaging. The Microscopy Services Laboratory in the Department of Pathology and Laboratory Medicine at UNC is supported in part by the P30 CA016086 Cancer Center Core Support Grant to the UNC Lineberger Comprehensive Cancer Center. The 16S rDNA sequencing was performed by the UNC Microbiome Core, which is overseen by the director Dr. Andrea Azcarate-Peril and is supported by the Center for Gastrointestinal Biology and Disease (CGIBD P30 DK034987) and the UNC Nutrition Obesity Research Center (NORC P30 DK056350). We would like to thank Monica Macharios for assisting with the Oil Red O analyses, Sarah Torzone for assisting with Oil Red O image acquisition, and Peter Breen and Sarah Torzone for assisting with data collection for the developmental timing experiments. Finally, we would like to thank Dr. Mark Peifer (UNC) and Dr. Bob Duronio (UNC) for critical reading of the manuscript.

## Author Contributions

**Conceptualization:** Rachel N. DuMez-Kornegay, Robert H. Dowen.

**Data curation:** Rachel N. DuMez-Kornegay, Lillian S. Baker, Alexis J. Morris, Whitney L. M. DeLoach.

**Formal analysis:** Rachel N. DuMez-Kornegay, Lillian S. Baker, Whitney L. M. DeLoach, Robert H. Dowen.

**Funding acquisition:** Rachel N. DuMez-Kornegay, Robert H. Dowen.

**Investigation:** Rachel N. DuMez-Kornegay, Robert H. Dowen.

**Project administration:** Robert H. Dowen.

**Supervision:** Rachel N. DuMez-Kornegay, Robert H. Dowen.

**Writing – original draft:** Rachel N. DuMez-Kornegay.

**Writing – review & editing:** Rachel N. DuMez-Kornegay, Robert H. Dowen.

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
