## [Decision Letter · Decision Letter 0]

15 Nov 2023

Dear Dr Dowen,

Thank you very much for submitting your Research Article entitled 'Kombucha Tea-associated Microbes Remodel Host Metabolic Pathways to Suppress Lipid Accumulation.' to PLOS Genetics.

The manuscript was fully evaluated at the editorial level and by independent peer reviewers. The reviewers appreciated the attention to an important topic but identified some concerns that we ask you address in a revised manuscript.

We therefore ask you to modify the manuscript according to the review recommendations. Your revisions should address the specific points made by each reviewer.

1) Provide a detailed list of your responses to the review comments and a description of the changes you have made in the manuscript.  Please note that several of comments from reviewers 1 and 2 only require modification of the text or toning down the interpretation of the results.  Please note the concerns raised by reviewer 3 that warrant detailed attention.

Yours sincerely,

Sean P. Curran

Guest Editor

PLOS Genetics

Gregory Barsh

Editor-in-Chief

PLOS Genetics

Reviewer's Responses to Questions

**Comments to the Authors:**

Reviewer #1: In this manuscript, DuMez-Kornegay et al. characterize the utility of a C. elegans model system in studying the impact of kombucha-tea associated microbes (KTMs) on organismal physiology. The manuscript is very well-written and extremely easy to follow, and the thoroughness of the experimental methodology and data analysis is commendable. The data is quite strong and well-controlled, and the model system is very interesting. I have very few comments about the study, all of which can be addressed textually. While I recommend a few experiments, these are all merely suggestions as the rigor of the data is sufficient that new experiments do not need to be performed, although in some cases (if the authors find it helpful) may benefit the strength of the manuscript.

-My only main concern of the study is that the authors very elegantly state in their introduction and conclusion that the benefit of producing this model is to better understand the impact of KTMs on physiology – something that is hard to do in humans. However, other than making very strong claims on the fact that KTMs do not impact overall organismal health, there is very little done to study the benefits of KTMs. Based on what the authors argue is the strength of making this model, this is a glaring weakness. Perhaps the authors could benefit by performing a few physiological assays to determine whether a KTM diet is actually beneficial to some readout of health: healthspan (e.g., motility, gut barrier integrity, etc.), lifespan, stress resilience, etc. I don’t expect the authors to do a comprehensive battery of physiological assays, so this can be minimally addressed by using the RNA-seq analysis to determine whether transcriptional profiles generally associated with increased health/longevity are found to be differentially expressed in their animals.

-The reproduction phenotype is sort of glossed over as being only “minor” effect, but it is still a statistically significant decrease. Generally, a decrease in brood size is considered a negative effect, yet the idea of KTMs is that there is some “health benefit”. The authors really need to address this. Perhaps try to determine why is there a decrease in brood: are the health of the progeny compromised due to the decrease in somatic fat (this can decrease available nutrients given to progeny). Is there a decrease in yolk transmitted to progeny (VIT-2::GFP)? Is this associated with the decrease in germline genes they see?

-The authors should at least comment in text about the decrease in desaturate expression, which is normally correlated with increase in specific fats, yet they see an overall decrease in fats (likely due to the increased expression of other lipases, etc.).

-Is it concerning that the authors see a decrease in neuron-related genes? Is neuronal function affected in these animals (e.g., behavior assays, chemotaxis, etc.)? Again, this can be addressed textually, and I don’t think actual experiments are necessary.

Reviewer #2: Dowen and colleagues present a research study that examines how microbes that are found in kombucha influence C. elegans. I commend the authors on an outstanding study that is well designed and with appropriate experimental controls. I found the manuscript clear to read and appreciate that the data was presented matter-of-factly and not over sold. In light of the popularity of kombucha to the general public, I believe this study will be of general interest beyond the scientific community. Moreover, this work expands upon the list of beneficial microbes that can influence organismal physiology. In my opinion, this work is a perfect fit for PLOS Genetics and can be accepted in its current form. I do note some VERY minor concerns that could help with readability, but these are not required and I leave it to the editor and author if they are helpful.

1. I would encourage the authors to discuss all data points that reach statistical significance in their work as the collectively might contribute to the overall outcomes. This does not require a significant addition to the discussion, but there are several weak phenotypes (but significant) that are not thoroughly accounted for and might be interesting.

2. It would be useful to better exploit the RNAseq data to identify classes of genes that have been found to impact organismal healthspan. These can be done with publicly available tools in WormBase.

Reviewer #3: In this manuscript by DuMez-Kornegay et al., the authors present the findings that consumption of the microbes found in Kombucha Tea, a popular fermented beverage with purported health benefits, by the nematode C. elegans leads to reduction in fat mass. The authors report that reductions in fat mass are accompanied by increases in lysosomal lipase expression and reduced expression of enzymes of fatty acid biosynthesis, leading them to speculate that lipid droplet catabolism by lipophagy is important for the observed changes in overall lipid stores, in parallel with reductions in lipid synthesis.

Major:

Pumping has been shown to be a poor corollary of food intake. At least the authors should acknowledge this; at best they could assess uptake of a vital dye such as C1-BODIPY-C12.

Because of the major intestinal colonization observed with KTM, the authors should test whether the intestine is involuted and/or the luminal volume is expanded. This could be consistent with an increase in autophagic activity and loss of lipid droplets. Also, is the intestine colonized by bacteria or yeast or both?

Pathogen exposure and genetic activation of the stress transcription factor skn-1 have been shown to dramatically reduce somatic fat, much like the effects observed here. The authors should test whether this is a pathogenic response as well as the dependency upon skn-1. The RNAseq data in figure 5 also indicate enrichment of a lipid and pathogen gene expression signature, making this possibility even more likely.

The authors should be sure statistical tests used are indicated in each legend, as well as statistical thresholds (e.g. missing in Fig. 1D).

The lipid quantification by oil red O indicates an extremely small difference in intensity (Y axis scaling should be from 0 not 6.20 to 6.40), whereas the differences seen by Nile red, biochemical quantification, or lipid droplet analysis indicate a much larger effect. The authors should explain the reason for the differences and perhaps consider or explain why the results are so quantitatively discordant.

C. elegans accumulates much more lipid as they transit reproductive life; given that the KTM animals appear smaller, is body size/growth an explanation for the lower fat mass observed? It does not appear to be alterations in time to adulthood based upon data in Fig. 3. The authors should look at later days of adulthood.

Lipid droplet number in addition to size should be included in the analysis in Fig 2/S3, if the analysis was completed properly to objectively quantify lipid droplet number per HPF, for example.

Filter sterilization may not be sufficient to remove all microbes/spores/bacterial products from the KTM-FM mix prior to introduction of the cultured bacteria. This should be addressed, experimentally if possible.

The “wash” experiments in fig 4H-I are informative but don’t entirely say that the small molecules are dispensible; rather they might be important for getting the bacteria to the state where they induce lipid depletion. This possibility should be indicated rather than dismissing the importance of these small molecules. Finally, are the authors certain that the post-fermented mix does not harbor other microbes that proliferated during the fermentation process? If filtration doesn’t completely sterilize the KTM-FM pre-fermentation mix, this could be a possibility. Sequencing could be used to rule this out.

The oil red o data in figure 6G show extremely blunted effect sizes, as above. A more sensitive method should be considered for assessing changes in fat, even though the effects are significant; a 6% difference is of dubious significance (and this appears to be the magnitude based upon the scaling of the axis. Why are the oil-red-o data showing compression of effect size if the effects are really as dramatic as indicated by the lipid droplet reporter and nile red?

The leap from lysosomal lipases to lipophagy is not substantiated and could be speculated in the discussion but should not be indicated in the results. If the authors want to substantiate this claim they should test the requirement of classic autophagy genes, as the effects on lipid levels could be indirect through increased classic oxidation or reduced synthesis. Experiments could be done to indicate increased oxidation (oxygen consumption) or decreased synthesis (e.g. 13C labeling as in Perez and Van Gilst PMID 18762027) but those may be beyond the scope of this investigation. Instead the authors could present the data on expression and functional significance and speculate in the discussion rather than declaratively concluding “lipophagy” as the abstract and results suggest. In contrast the DGAT OE experiment is more convincing, but still the effect size is very small again because of the use of the oil red o assay.

Minor:

Introduction--the authors may want to cite direct consequences of reducing microbial diversity, e.g. C. difficile colitis in response to antibiotics.

Line 61 “hepatoprotective” is not a widely used term, the authors may want to expound on the meaning they intend.

Rather than “standard C. elegans husbandry” the authors may wish to use “Axenic preparation of C. elegans cultures renders…”

The authors may want to specify that NGM-KTM plates contained no antibiotics or antifungals.

Line 224, 225, 228 ab libitum should be ad libitum.

It looks like KTM could disrupt the pulsatility of mlt-10 expression/turnover. Also the KTM data for 3A/B/C appear to be the same without replication and without statistics. How many times was this experiment done and were the results representative? How many animals were counted at each timepoint?

The authors could consider a chi-squared statistic on Figure 3G to show significance.

Did the authors expect to see changes in developmental genes at adulthood (Fig. 3D-F)? Why was an earlier timepoint not queried if differences in development were to be uncovered?

The statement on lines 291-294 is not well justified by the 5xKTM experiment and should be altered or removed.

**Have all data underlying the figures and results presented in the manuscript been provided?**

Reviewer #1: Yes

Reviewer #2: Yes

Reviewer #3: Yes

PLOS authors have the option to publish the peer review history of their article (what does this mean?). If published, this will include your full peer review and any attached files.

Reviewer #1: No

Reviewer #2: No

Reviewer #3: No

---

## [Decision Letter · Decision Letter 1]

13 Feb 2024

Dear Dr Dowen,

Thank you very much for submitting your Research Article entitled 'Kombucha Tea-associated Microbes Remodel Host Metabolic Pathways to Suppress Lipid Accumulation.' to PLOS Genetics.

The manuscript was fully evaluated at the editorial level and by independent peer reviewers. All reviewers agree that the authors have addressed the previous concerns. Reviewer 3 has requested some minor corrections to the text, which I believe should be addressed before formal acceptance.  

We therefore ask you to modify the manuscript according to the review recommendations. Your revisions should address the specific points made by reviewer 3.

Yours sincerely,

Sean P. Curran

Guest Editor

PLOS Genetics

Gregory Barsh

Editor-in-Chief

PLOS Genetics

All reviewers agree that the authors have addressed the previous concerns. Reviewer 3 has requested a some minor corrections to the text, which I believe should be addressed before formal acceptance.

Reviewer's Responses to Questions

**Comments to the Authors:**

Reviewer #1: The authors have done a commendable job in addressing all the comments of the reviewers. The revision is highly improved and this manuscript is a fantastic candidate for PLoS Genetics.

Reviewer #2: The authors have addressed my concerns.

Reviewer #3: The authors for the most part have satisfactorily addressed my concerns. Only a few very minor points and/or errors:

BODIPY staining except at high doses mostly stains gut granules. Many references show this e.g. Yen et al., PlosOne 2020, so the authors may want to adjust the statement 346-352 to indicate that acidified organelles and lipid droplets are illuminated by the BODIPY feeding.

The legend for figure 4J has an error, it does not show images of Calcofluor, and there are 2 “K” references.

Strictly speaking the t-test is not an appropriate statistical test for figure 6A-E and 6J unless it is multiple hypothesis testing corrected. Meanwhile 6K cannot be one way anova because it has only 2 datasets. I suggest the authors look very carefully through their figure assignments and legends one last time.

**Have all data underlying the figures and results presented in the manuscript been provided?**

Reviewer #1: Yes

Reviewer #2: Yes

Reviewer #3: Yes

PLOS authors have the option to publish the peer review history of their article (what does this mean?). If published, this will include your full peer review and any attached files.

Reviewer #1: **Yes: **Ryo Higuchi-Sanabria

Reviewer #2: No

Reviewer #3: No

---

## [Editor Report · Decision Letter 2]

22 Feb 2024

Dear Dr Dowen,

We are pleased to inform you that your manuscript entitled "Kombucha Tea-associated Microbes Remodel Host Metabolic Pathways to Suppress Lipid Accumulation." has been editorially accepted for publication in PLOS Genetics. Congratulations!

Yours sincerely,

Sean P. Curran

Guest Editor

PLOS Genetics

Gregory Barsh

Editor-in-Chief

PLOS Genetics

Comments from the reviewers (if applicable):

**Data Deposition**

http://datadryad.org/submit?journalID=pgenetics&manu=PGENETICS-D-23-01118R2

**Press Queries**

---

## [Editor Report · Acceptance letter]

6 Mar 2024

PGENETICS-D-23-01118R2 

Kombucha Tea-associated Microbes Remodel Host Metabolic Pathways to Suppress Lipid Accumulation. 

Dear Dr Dowen, 

We are pleased to inform you that your manuscript entitled "Kombucha Tea-associated Microbes Remodel Host Metabolic Pathways to Suppress Lipid Accumulation." has been formally accepted for publication in PLOS Genetics! Your manuscript is now with our production department and you will be notified of the publication date in due course.

With kind regards,

Anita Estes

PLOS Genetics

On behalf of:
